# Flow parsing as causal source separation allows fast and parallel object and self-motion estimation
Malte Scherff ✉ & Markus Lappe

Optic flow, the retinal pattern of motion experienced during self-motion, contains information about one's direction of heading. The global pattern due to self-motion is locally confounded when moving objects are present, and the flow is the sum of components due to the different causal sources. Nonetheless, humans can accurately retrieve information from such flow, including the direction of heading and the scene-relative motion of an object. Flow parsing is a process speculated to allow the brain's sensitivity to optic flow to separate the causal sources of retinal motion in information due to self-motion and information due to object motion. In a computational model that retrieves object and self-motion information from optic flow, we implemented flow parsing based on heading likelihood maps, whose distributions indicate the consistency of parts of the flow with self-motion. This allows for concurrent estimation of heading, detecting and localizing a moving object, and estimating its scene-relative motion. We developed a paradigm that allows the model to perform all these estimations while systematically varying the object's contribution to the flow field. Simulations of that paradigm show that the model replicates many aspects of human performance, including the dependence of heading estimation on object speed and direction.

Optic flow is the pattern of light on the retina that results from relative motion between an observer and their visible surroundings. It provides valuable information about the structural layout of the scene, including its rigidity, and the different types of motion present[1,2]. Recovering self-motion information from optic flow is not only theoretically possible but also crucial for safe navigation in daily life[3–5]. Likewise, detecting an independently moving object[6,7], estimating its trajectory[8–11] and determining its time to contact[12,13] during self-motion is evident when considering simple examples in sports. For instance, a soccer player must steer their movement towards a position on the pitch where the ball will likely land to receive an inaccurate pass from a teammate successfully.

However, it remains unclear how the two processes—estimating self-movement and deriving information about sources of independent motion - are connected and interact when they are based on the same flow information, and how the causal attribution of any motion to self-motion or object motion is achieved.

## Theoretical considerations and psychophysical evidence for the estimation of self-movement

The simplest form of flow patterns is one in which everything moves radially away from a singular point, the focus of expansion (FOE). Such a radial flow field occurs when an observer moves through a stationary scene, and the movement solely consists of a translation in a certain direction. In that case, the direction of self-motion, and the FOE coincide. It is well established that humans can recover the direction of motion from such flow patterns with an accuracy well within a range of 1 to 2 degrees of error[3,4,14].

While such radial patterns are often used in psychophysical studies, it is well known that eye-movements that occur during self-motion confound this simple structure by adding rotational components[15–19]. Nonetheless, studies in which participants performed eye movements showed that self-motion estimation is still possible with an error between 2 and 4 degrees[16,20–22]. Even when the eye movements were simulated, some studies found heading estimation performances of similar[16] or slightly worse quality[22] with heading errors up to 5 degrees. Further studies have shown that self-motion estimation is quite robust such that adding noise to flow stimuli reduces but does not impede the ability to estimate the direction of self-motion[23,24].

Studies have also shown that the presence of an independently moving object (IMO) in the optic flow can bias heading estimation[25–28]. In some of those studies, the objects in the scene moved such that the combined flow only consisted of lateral motion. By moving the object backward the same amount the observer moved forward the relative motion between IMO and the observer was only due to the object movement. Hence, the combined flow of the object contained no information about the heading direction.

Department of Psychology & Otto Creutzfeldt Center for Cognitive and Behavioral Neuroscience, University of Münster, Münster, Germany.
✉e-mail: malte.scherff@uni-muenster.de

Heading estimation was then biased in the direction of the object movement[26,28]. Otherwise, when there was a change in the relative depth between object and observer, the bias was in the opposite direction of the object motion[25,27–29]. Another observation about the influence of object movement on heading estimation was made in a study in which the object's speed was systematically varied[30]. Increasing the speed of the object from zero first led to an increase in heading estimation error, which peaked for the intermediate speeds tested, and then reduced the object's impact on the heading estimation as the error dropped to nearly zero for further increases of object speed.

## Theoretical considerations and psychophysical evidence for the estimation of independent object motion during self-motion

Gaining information about an object while moving from optic flow alone presents a challenge, as no part of the visual field exclusively contains image motion caused by the object's movement. Self-motion always confounds it. Nonetheless, humans are able to detect an independent source of motion in an optic flow stimulus based on the divergence in direction[6] or speed[7,31] to the rest of the pattern. When participants were tasked with judging the trajectory of a moving probe in a radial flow field, the results were consistent with an interpretation in a world-relative reference frame[8–11]. These results align with the idea of a process called *flow parsing* with which the brain separates retinal motion components due to self-motion from those due to object motion. While no explicit mechanism was proposed, this might be achieved by using the brain's selectivity to flow patterns (e.g., ref. [32]) to identify and globally discount the component due to the self-movement. The remaining flow could then be attributed to an independent source of motion and used to gain information about it. This might amount to an iterative solution, in which self-movement estimation (possibly biased) is followed by determining object movement, which relies on the prior estimate of self-movement, and then the process may be iterated to refine each solution[27,33].

However, some studies have, seemingly paradoxically, suggested that estimation of independent object motion does not rely on prior estimation of self-motion. For example, Warren et al. showed that participants' judgment of an object's trajectory was not consistent with its scene-relative movement when self-motion perception was strategically biased and proposed that flow parsing might occur independently of heading estimation[34]. Further evidence for that was provided by Rushton et al. as they found performance in object movement estimation to be more precise than in heading estimation[35]. Therefore, it is unlikely that the identification of scene-relative object movement relies on the prior self-motion estimation, especially as another study suggests that such identification might solely be driven by optic flow processing[36]. These findings suggest that self-motion and object motion estimations might share initial processing stages but might ultimately take place in parallel and independent of each other.

## Computational models for heading and object estimation

One of the first models of heading perception was the population heading map model of Lappe and Rauschecker[37,38]. It is based on the subspace algorithm of Heeger and Jepson, which recovers self-motion parameters under the assumption of a rigid scene[39]. Heeger and Jepson tested their model on artificial flow fields derived from simulated 3D scenes and realistic ones from real-world camera recordings. Compared to other heading estimation methods available at the time, their model performed well and recovered translation even when varying degrees of uniform noise were present in the flow. Potential biases due to object motion would result from systematic mis-estimation of self-motion. Sauer et al. used the subspace algorithm to estimate heading for simulations of distorted optic flow in ophthalmic correction lenses[40]. The strength of the distortion effect on the flow varied depending on where the gaze crossed the lens. The resulting heading estimation bias was similar to that reported by novice wearers of progressive adaptive lenses.

Lappe and Rauschecker devised a biologically plausible implementation of the subspace algorithm in a two-layer neural network of heading perception[37,38]. The input layer, designed after monkey area MT, contains neurons that encode motion direction and speed, which results in a population-encoded representation of the optic flow. Neurons in the second layer are selective for directions of translational ego-motion and model the next stage of the flow processing pathway, area MSTd[41]. The activities of populations of these neurons form a retinotopic heading map in which the activity peak indicates the direction most consistent with the input flow. To achieve this output, the connection strengths between the neurons of the two layers are computed using the subspace algorithm. Heading estimation performance was tested across a variety of settings with this model[5,37,42–44]. Results were well in line with human behavior as long as the input flow was dense enough, i.e., it consisted of at least 10 points. When the 3D scene was sufficiently non-planar, or the level of uniform noise added was low enough, even simulated eye-movements did not significantly interfere with the heading estimation. Additionally, the model could reproduce the mis-localization of the FOE under the optic flow illusion[45]. Later results showed that the model also reproduces the heading biases seen in human observers when the flow field contains an independently moving object[28].

Another early model concerned with a bias introduced by independent object motion is the motion pooling model developed by Warren and Saunders[25]. Their two-layer model, which uses template matching to estimate the direction of self-motion, successfully replicated the heading estimation bias induced by approaching objects and explained it as the average of the FOEs due to observer and object motion. However, subsequent tests showed that it was not suitable for explaining the change in bias direction for purely lateral combined flow[46]. Layton et al. followed a similar modeling approach, early motion pooling combined with template matching, but additionally equipped with competitive dynamics between the matching cells[47]. While self-motion parameters were not estimated, heading biases were found in shifts of activity peaks of heading maps. These dynamics allowed to explain the change of heading bias direction for different types of object motion. Extending this model to include recurrent connections allowed it to capture the temporal dynamics of self-motion estimation in the presence of an IMO[46].

Another type of heading estimation model was inspired by the early work of Longuet-Higgins and Prazdny[2] and Rieger and Lawton[48]. Their analysis of flow fields with rotational components showed that local differences in flow vectors, given a sufficient difference in depth, can provide a radial flow field for which the FOE coincides with the true heading direction. Hildreth used this method in a computational heading model so that it could deal with small, independently moving objects[49]. By taking the direction that agrees with the majority of flow differences in different regions of the visual field as the estimated heading, small objects that caused inconsistent flow were omitted. Royden later adopted this idea and used motion-opponent operators inspired by neurons in primate area MT to implement this motion subtraction[50]. Assuming a rigid scene apart from observer motion, heading was estimated by comparing maximally responding operators to translational heading templates. This model was later improved to deal with moving objects by adding Gaussian weighting to the connections between the operators of the first layer and the template cells[51]. Interestingly, the author states that this addition was necessary to remove biases caused by objects away from the FOE, making this model less suitable for some of the newer data[28]. Another model extension resulted in one of the few optic flow-based models for object detection[52]. After heading estimation, the first layer operators' preferred directions and response magnitudes were compared to the template that determined the estimate. If the direction differed too much or the response magnitude was significantly higher than the responses of other cells, it was assumed that there was a self-moving object in the scene with a boundary at the operator's location. While this model showed promising and robust results under various circumstances, it used heading estimation as a prerequisite for the object detection process.

The approach Raudies and Neumann used in their computational study to estimate self-motion from optic flow containing a moving object was different[53]. Their analytic model relied on local segmentation cues such as accretion/deletion, expansion/contraction, and acceleration/deceleration

to qualitatively reproduce the behavioral pattern of heading bias. This, however, is not in line with the results of Li et al. who showed that the heading estimation process does not include the segmentation of independently moving objects[28].

Compared to optic flow based heading estimation models, the field of object motion estimation models is sparsely populated. Layton and Fajen presented a neurophysiologically inspired model of object motion recovery during self-motion[27]. It uses interactions in MT and feedback from MSTd to MT to transform retinal object motion into a world-relative reference frame. A key prediction was that such a process, which shifts initial MT responses reflecting the retinal motion pattern to align with world-relative motion, depends on a temporal process in MT. A more elaborate version of the model has been developed by Layton and Niehorster[54]. It uses two separate processing streams, one for self-motion estimation consisting of MT cells with reinforcing surrounds projecting to MSTd, and the other using MT cells with suppressive surrounds connected to ventral MST, to estimate scene-relative object motion. The latter pathway is modulated by the estimates of the former. While recent findings indicate that MT activity is modulated in accordance with scene-relative object motion[55], it remains open whether the reported time course matches the time course of real life situations, in which eye movements frequently disrupt the retinal flow such that only segments of a duration of about 300ms are available for processing[19].

### Aim of the study

We present a computational model that processes optic flow and can estimate self-motion direction and parameters of an independent source of motion in parallel. To do so, the model uses flow parsing to separate information from different causal sources of flow. The flow parsing process is based on likelihood maps computed for different parts of the flow field. These maps indicate the consistency of the corresponding flow with various heading directions. Depending on that consistency, the likelihood maps are used for either heading or object estimation. The model's structure allows these estimations to be run in parallel without needing recurrent or feedback connections.

When comparing the model's performance for a self-motion scene, including an IMO, with research from studies in the literature, the results align with behavioral data. To be more specific, the model's heading estimation process gives rise to an error that systematically depends on object speed, with slow and fast moving objects causing a small error and an error peak for intermediate speeds, similar to findings of Dokka et al.[30]. Additionally, the model's direction of mis-estimation depends on whether the object maintains a fixed depth relative to the observer, a finding reported in various studies[25–29]. The model's object detection performance depends on the deviation of the object flow from the background pattern, as Royden and colleagues found[6,7]. While there is no research providing behavioral data in regard to object localization in optic flow fields, the model is able to successfully localize the independent source of motion solely based on flow velocities. Lastly, the object direction estimation is similar to human performance that shows that the perceived trajectory is consistent with scene-relative motion[9–11].

## Results
### The FLOW PARSE model

The *FLOW PARSE* (FLOW-based Parallel Source Estimation for self- and object motion) model is designed to perform three tasks: infer the causal sources of retinal motion, estimate self-motion, and estimate the parameters of an independently moving object. It implements the subspace algorithm[39] for computing heading likelihood maps, so-called residual surfaces, that indicate the consistency of optic flow with a grid of heading directions. The shape of these surfaces can indicate the causal sources of the corresponding flow. Analysis of these shapes is the foundation of the flow parsing process. It regulates the inputs to the self-motion and object estimation pathways of the model to minimize the use of information from flow not relevant to the respective tasks. This multi-layered model only contains forward connections between the layers, so any outcome influences neither process. After describing the structure of the residual surfaces and the model, we will present simulation results for a self-motion scenario that allows for the simultaneous estimation of various scene parameters. We then compare these results to human performance from studies with comparable paradigms.

### Structure of residual surfaces

The residual surfaces computed with the subspace algorithm can be seen as heading likelihood maps, as they indicate the consistency of various heading directions with a given optic flow field (see "Methods" Section for details). Earlier models that used this method for heading estimation took a similar approach in handling residual surfaces[37,39,40]. Given a flow field, they were computed for various parts of the visual field and summed up as a heading map where the peak indicates the best solution. The corresponding studies have not analyzed the distributions of the residual values in the separate surfaces and whether further information could be inferred. Hence, only some of those distributions were reported for illustration purposes. While Heeger and Jepson showed a residual surface with two peaks[39], indicating multiple distinct translation directions as potential solutions, they clarified that this resulted from a known ambiguity of the flow field in self-motion towards a single plane[2,15,56]. Apart from that, only surfaces with a single peak were reported, and our implementation confirmed this when we examined similar flow fields and the resulting residual distributions. This was true for all flow fields with self-motion through a rigid environment. However, we often found multiple peaks found when the scene includes an independent source of motion.

Figure 1 shows residual distributions for three flow fields. The first distribution results from flow solely due to observer translation in an environment with a random depth distribution. Hence, the flow field is rather uniquely solvable, and the residual surface shows a single peak with a smooth descent. The structure of the residual distributions changes when object motion is introduced into that scene, as seen in the second panel. There, the flow field contains an object placed to the right of the FOE. The object moves such that the combined flow is lateral to the right and mostly in line with the overall flow pattern. In terms of speed, the combined flow is faster than the observer flow. This results in a residual surface in which a second, albeit smaller, peak emerges compared to the first surface. Taking this residual surface as a likelihood map for heading, multiple distinct directions could be seen as solutions for the flow field. However, the higher peak indicates a more likely solution. Due to being a local minimum for the direction orthogonal to the peaks, the area between the peaks resembles a saddle point whose position coincides with the object's location. The last panel shows an example of a residual surface for when the combined flow is faster than the surrounding observer flow and deviates in direction from the flow pattern. The combination of observer and object movement gives rise to purely vertical combined flow, leading to a residual surface with a distinct saddle point, again matching the object's location. This time, the saddle point's orientation, the axis on which the peaks emerge, does not entirely align with the combined flow direction but is slightly tilted relative to vertical. Neither peak emerged at a location close to the true heading direction.

Overall, the residual structures carry information about the independent source of motion in the scene. The retinal location of the object is consistently indicated by the saddle point between the residual peaks, and its orientation of the peak placements varies depending on the combined flow direction.

### Structure of the model

The model consists of multiple layers and in most of the layers, we employ a specific type of operator to process the input the layer receives. All types of operator are at least loosely inspired by certain types of neurons and their capabilities as well as earlier implementations of the subspace algorithm. As input from and output provided by each layer are represented as retinotopic maps and the range of each operator is restricted to a certain area of these maps, we refer to these areas as the receptive fields of the operators. The general structure of

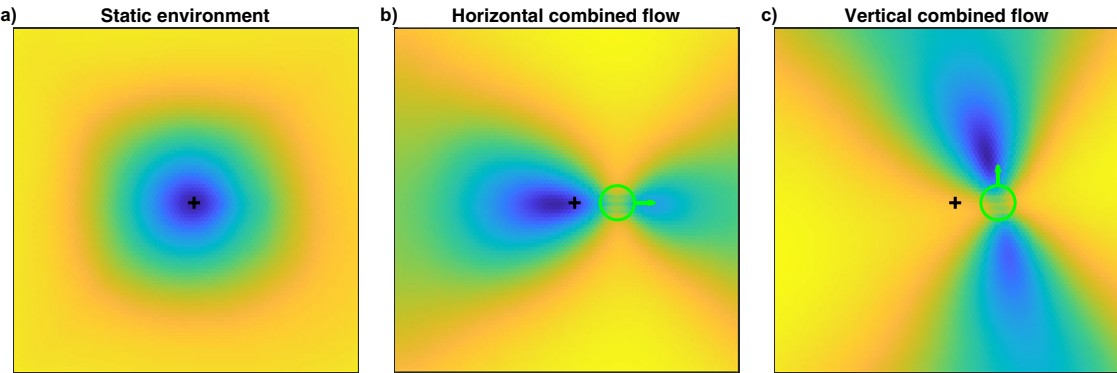

**Fig. 1 | Examples of residual surfaces.** Corresponding flow fields are due to simulated observer translation towards a static cloud of dots. The black plus indicates the FOE of the radial flow pattern and, therefore, the heading direction. The object, if present in the scene, is depicted with a green circle, and the green arrow indicates the direction of the combined flow. **a** The residual surface has a singular peak, which coincides with the true heading direction when no object is present. **b, c** When present, a moving object gives rise to residual surfaces with a saddle point at the object's location, and the position of the surrounding peaks changes with the object's flow direction.

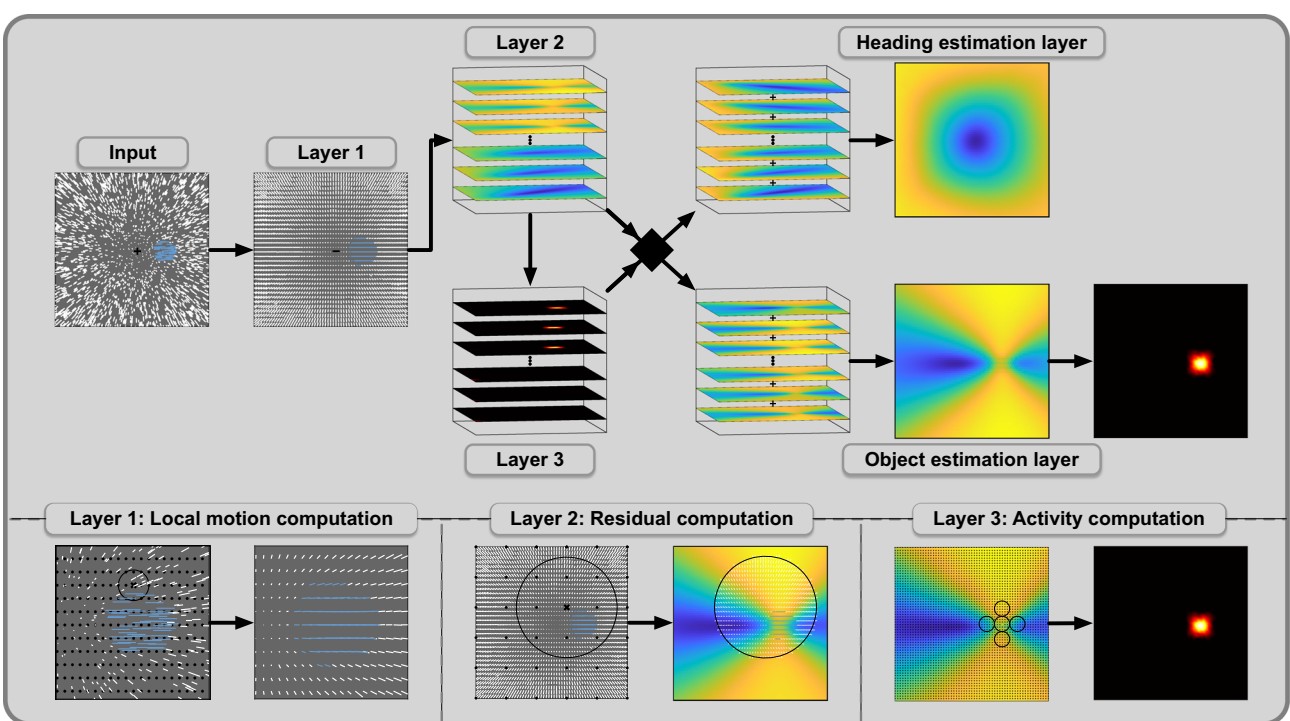

**Fig. 2 | Structure of the computational model.** The model processes optic flow to detect the presence of a moving object, estimate self-motion direction, estimate object motion direction, and determine object location. It consists of multiple layers with specific processing operators in each layer. Optic flow fields form the input into the model. White and blue vectors represent observer flow and combined flow, respectively. Layer 1 computes a vector representation of the input flow by computing local averages of speed and direction. It contains operator units with small circular receptive fields that cover the FOV. The black circle shows the receptive field of one operator as an example, and the locations of the remaining operators are indicated with black dots. Layer 2 computes residual surfaces for flow in different parts of the visual field. Flow vectors in the receptive field of one operator are grouped, and one residual surface is computed for each group. Layer 3 computes activity maps, one for each residual surface. Operators calculate average residual values in parts of their cross-shaped receptive fields and use differences between them to compute activity. Receptive fields vary in size and orientation and cover the residual surface. The flow parsing process is based on the activity maps and decides whether a residual surface is used for heading or object estimation. The corresponding surface is used for object estimation when the activity maximum is high enough. The heading estimate is determined by the peak of the heading map, which is the result of summing the respective surfaces. For the object estimation, the incoming surfaces are summed up, and an activity map is computed. Size and position of the activity maximum determine object detection and localization, respectively. To estimate the object's movement direction, the orientation of the operators that contributed to the activity is used.

the model, as well as a more detailed presentation of all layers, can be seen in Fig. 2.

The goal of the first layer is to create a vector representation of the incoming optic flow by averaging the flow at different retinal locations[43,57]. Based on neurons found in the middle temporal visual area (MT) that show selectivity for direction and speed[58,59], we place the operators of this layer so that the field of view (FOV) is covered by the corresponding receptive fields in which the mean speed and direction of flow are computed. This results in a vector representation of the optic flow across the FOV.

Area MT projects to the dorsomedial region of the medial superior temporal area (MSTd) that contains neurons showing heading-sensitivity based on larger flow patterns typical for self-motion[32]. Hence, the second

layer aims to test the consistency of flow from different retinal regions with the candidates in the heading space. The retinal regions are defined by the receptive fields of the operators corresponding to this layer that cover the visual field. Flow vectors in the same receptive field are grouped, and residual functions defined by the subspace algorithm are evaluated on each group. This results in residual surfaces, one for each of the groups. Therefore, a single flow field gives rise to several residual surfaces, one for each second layer operator. Each residual surface contains values that indicate the likelihood of the candidate directions to have caused the part of the flow field in the respective retinal region.

The third layer plays a crucial role in the model, serving as the foundation for the flow parsing process. It assesses the distribution of the residual values since a distinct saddle point indicates the presence of a source of independent motion in the corresponding retinal region. Containing one, such surfaces will be used for the object estimation process, while single peak surfaces will contribute to the heading estimation. Therefore, the set of surfaces is parsed due to inferred causal sources of motion and then channeled accordingly to the next processes.

In order to find a saddle point on a residual surface, we implemented saddle point operators designed to show activity when they are placed close to a saddle point area of a certain size and orientation. For that, the receptive field of such an operator consists of 5 circular same-sized areas arranged in a cross-shape. The operator averages the residual values in each area and compares the surrounding values to the central one. Operator activity is either the sum of absolute values of the differences between central and surrounding values or zero if the signs of those differences do not alternate. Hence, for an operator with non-zero activity, there are exactly two directions from its central towards higher-valued areas, which we will call its peakward directions. In a multi-scale approach, we group saddle point operators that vary in the size of the circular areas of their receptive fields and their orientation and employ them throughout the incoming residual surface, which we transform by applying the negative logarithm and scaling the result to the range from 0 to 1. Summing the activity of each operator in such a group gives a cumulative activity for the retinal location at which that group was employed. This results in a retinotopically organized activity map that shows increased activity where the corresponding residual surface has a saddle point. An activity map with no or low activity indicates that the corresponding residual surface is due to observer flow only. Thus, if the activity map fails to exceed a certain *activity threshold* $\tau_1$, the residual surface will be assigned to the heading estimation layer or else to the object estimation layer.

The heading estimation layer sums all incoming surfaces, which results in the heading map. The peak in that map then indicates the estimated translational heading direction to explain the given flow field. As no further processing is needed beforehand, no operator type is employed for this layer.

The object estimation layer likewise aggregates all surfaces it receives. Then, to identify the location and properties of the object, the saddle-point operators are applied again on the aggregated map. If enough activity is detected, that is, if the activity maximum surpasses the second activity threshold $\tau_2$, the model assumes the presence of an independent source of motion in the flow field. In that case, the retinal location of the maximum saddle-point activity is then declared as the estimated location of the object. The object direction estimate is a weighted average of peakward directions of the saddle point operators at the object's estimated location. For every operator that contributes to the activity maximum, the direction with the smaller angle to the combined flow direction is used, weighted by the operator's activity. By restricting the selection of the peakward directions to those more similar to the combined flow direction, we avoid opposing directions canceling out their contribution to the estimation. Additionally, the largest angle possible between the estimated direction and the combined flow direction is 90°.

## Simulation results

The simulated paradigm provided flow fields based on a complex scene that contains observer and object movement. Due to the flow field's additivity, the flow presented to the model, the combined flow, is the sum of two components due to either observer or object movement, the observer flow and the object flow, respectively. While observer movement was a simple forward translation, object movement in the world combined horizontal movement of varying speed with an in-depth movement that we varied in different motion conditions and that was based on the observer translation (see Fig. 3 for the illustration of flow components and Fig. 9 for the detailed paradigm description). Simulation results are presented either by motion condition and horizontal object movement speed or as a function of speed ratio and directional deviation, metrics that indicate how the average flow speed or direction changes due to the addition of the object flow (Fig. 3).

**Setting activity threshold**. The main feature distinguishing our model from previous implementations of the subspace algorithm is that not all computed residual surfaces are used for heading estimation. Those residual surfaces are used to extract information about the independent source of motion. The process that decides for which a residual surface is used depends on its distribution and whether the maximum of the corresponding activity map surpasses the activity threshold $\tau_1$. Hence, the choice of $\tau_1$ regulates the flow parsing quality and, therefore, the input on which those processes depend. Thus, the first step in testing the model is to choose an appropriate threshold. In ideal circumstances for estimating self-motion, in a static scene with depth variation and enough points for dense flow fields, residual surfaces provide redundant information, and most of them should be used for heading estimation. The threshold $\tau_1$ should be chosen to exploit that redundancy. As the structures of residual surfaces in a more complex situation, perhaps more noisy or containing independently moving objects, tend to give rise to more variable activity, fewer surfaces would be assigned to the heading estimation layer. Due to the redundancy mentioned above, there should still be enough information to estimate self-motion reliably in such cases.

To determine a value for $\tau_1$, we started with a simulation in which the environment is entirely rigid. With no moving object in the scene, all flow is solely due to observer movement. The resulting residual surfaces are, therefore, all eligible for heading estimation. Figure 4a shows how the average rate of residuals used for heading estimation depends on the choice of the activity threshold. We want to adjust the flow parsing quality so that ~90% of surfaces enter the heading estimation layer, which we achieve by setting the activity threshold to $\tau_1 = 3$.

However, 90% is not a critical value, as preliminary testing showed that activity thresholds corresponding to values between 40% and just barely 100% give rise to similar estimation patterns to those that will be presented further below, although the magnitude of the patterns varied.

**Flow parsing**. For the main simulations, we introduce an independent source of motion into the scene so that some residuals are based on flow partly due to object movement and should be used for object estimation. Figure 4b shows how often residuals end up in the correct layer and how this depends on the speed of the object's horizontal movement and the motion condition. A maximum rate of around 90.8% is reached for all motion conditions when the object moves the fastest at 1 m/s. On the other hand, the slowest horizontal object speed (0 m/s) yields low rates of 61.8%−62.8% for motion conditions in which the object is not approaching the observer. Otherwise, the residuals are correctly assigned at 74.6% and 83.2%, respectively. For all motion conditions, increasing object speed lowers the chance of residuals being wrongly assigned for heading estimation.

The relation between flow parsing quality, speed ratio, and direction deviation can be seen in Fig. 6a. For flow fields that generated the lowest flow parsing quality, only 30.5% of residual surfaces were correctly assigned to the correct layer. In all these cases, the combined flow was either slower than the observer flow, so with a speed ratio below 1, or had a mean directional deviation below 10°. In general, the flow parsing quality increases when the combined flow is either faster or shows a higher deviation in direction. This is especially apparent when focussing

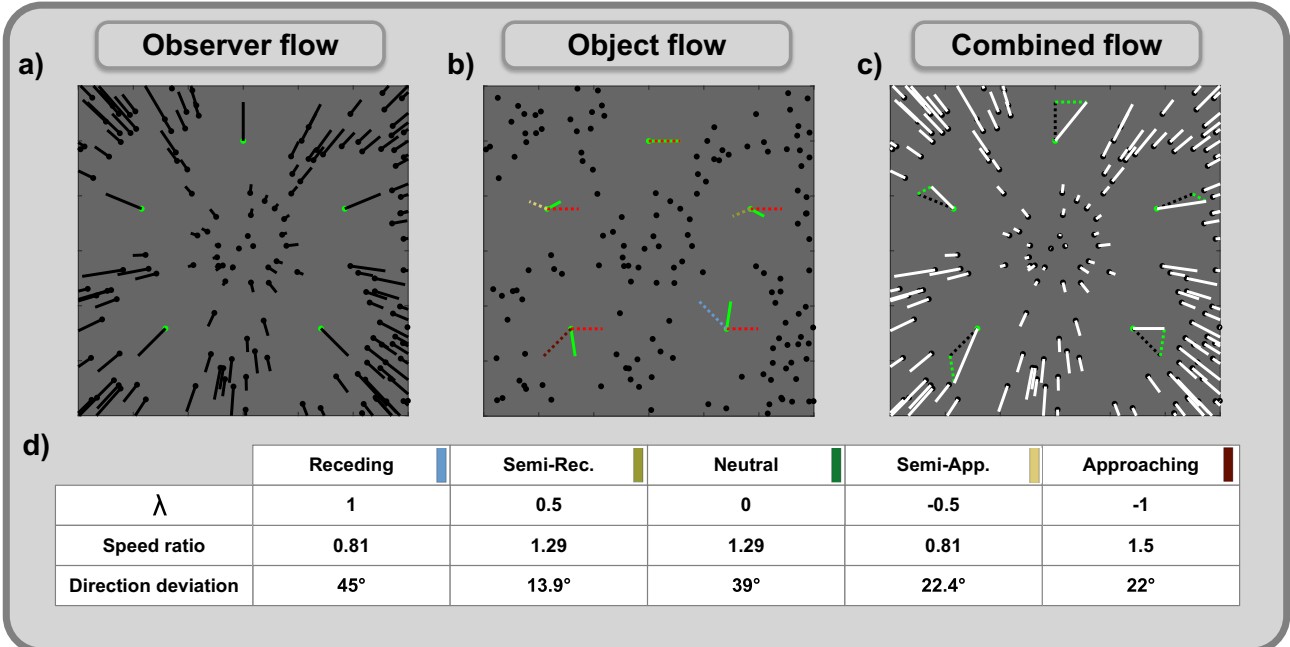

**Fig. 3 | Combination of observer and object motion and the resulting flow components used for the simulations.** Black points indicate the static points of the scene, and green ones illustrate five cases of independently moving objects. Objects, for clarity represented by only one point each, are placed at five of the 49 retinal positions used in the study, each serving as an example for one of the five object motion conditions. The paradigm simulated to validate the model always actually contained only one object per scene, which consisted of multiple dots (see Fig. 9 for the detailed description). **a** shows the observer flow (black lines), which occurs when only the observer moves. Observer movement is a translation towards the center of the panel, resulting in a radial pattern encompassing all points in the scene. **b** shows the isolated flow of the objects. Object movement in the world consists of two components: the horizontal movement resulting in horizontal flow (red, dotted lines) and the component based on motion in depth, both the observer's (translation $T$) and the object's, which is varied in the different conditions (dotted lines, colors corresponding to conditions specified in (**d**)). The flow resulting from the object's motion in depth is a multiple of the observer flow as the object moves in the direction $\lambda \cdot T$ in the world. The object flow (green lines) is the sum of the flow due to the horizontal and the in-depth movement (dotted lines). **c** shows the combined flow (white lines), which is sum of observer and object flow. **d** shows two flow metrics resulting from adding object movement into a previously static scene. This addition changes the flow in speed and direction of all points related to the moving object. Speed ratio indicates the rate of average velocities of combined flow to observer flow, with values above 1 signaling an increase in flow velocity. Direction deviation is the average, unsigned angle between the combined and observer flow vectors.

on flow fields where combined flow only deviates in speed or direction (red dashed lines in Fig. 6a). On average, 66.4% of residuals are correctly assigned for directional deviation below 10° and which rises to 83.8% when the average deviation is 45°. Similarly, average flow parsing quality is at 67.1% for speed ratios below 1 and at 82.8% when the speed ratio reaches 2. This shows that the average flow parsing quality increases as the combined flow deviates from the pattern.

### Heading estimation

Heading error. In order to estimate the direction of self-motion from an optic flow field, the model locates the peak of the heading map, which is the result of summing all residual maps assigned to the heading estimation layer. Figure 5a shows how close the estimation is to the actual parameter. When no self-moving object is in the scene, the heading error averages to 0.42 degrees of visual angle (dva).

An independent source of motion in the scene disturbs the flow pattern due to observer movement. Thus, parts of the flow are no longer valid cues for self-motion. When such an object is present in our simulation, the heading error follows a similar pattern regarding object speed for all motion conditions. When the object is not moving horizontally, the error is similar to baseline, ranging between 0.41 dva and 0.62 dva. It peaks at values between 0.88 dva and 1.56 dva for intermediate speeds before decreasing to 0.54 dva for the fastest objects. The heading error is largest for motion conditions where the object recedes.

Only a small error can be seen for flow fields where the addition of object movement only caused a noticeable change in flow velocity (Fig. 6b). In contrast, when the speed ratio was close to 1, the average heading error peaks at a direction deviation of 22° at 1.4 dva. The largest heading error of

6.8 dva was found for a flow field in which the combined flow was as fast as the observer flow at that location but deviated in direction by 48.5°.

Heading bias. To further characterize the systematic of the heading error, we calculated the extent of the mis-estimations in two particular directions: First, in the direction of the object's movement, and second, in the direction towards its location. These potential biases can be seen in Fig. 5b and Fig. 5c. As the magnitude of a heading bias is limited by the size of the heading error, we focus on the range of object speeds yielding the largest mis-estimations, i.e., between 0.125 m/s and 0.5 m/s. When the object recedes, heading estimation is biased in the object's direction for up to 1.15 dva. For the other conditions, heading error is biased in the opposite direction of the object's movement, peaking at 0.73 dva. For the same range of speeds, we found a peak bias value of 0.34 dva towards and 0.17 dva away from the object's location.

Comparing the peak bias values to the respective heading error, we can see that bias regarding object movement direction accounts for up to 80% of the heading mis-estimation. At the same time, bias towards or away from the location covers a maximum of 23%.

### Object estimation

Object detection. Residual surfaces are assigned to the object estimation layer when they sport a saddle point pronounced enough that the maximum of the corresponding activity map exceeds $\tau_1$. The model assumes the presence of an IMO when the activity map we compute for the result of summing all of those incoming surfaces has a maximum that passes a second threshold $\tau_2$. By our choice of $\tau_1$, around 10% of residuals create enough activity to be misused for object estimation. Hence, based on the idea

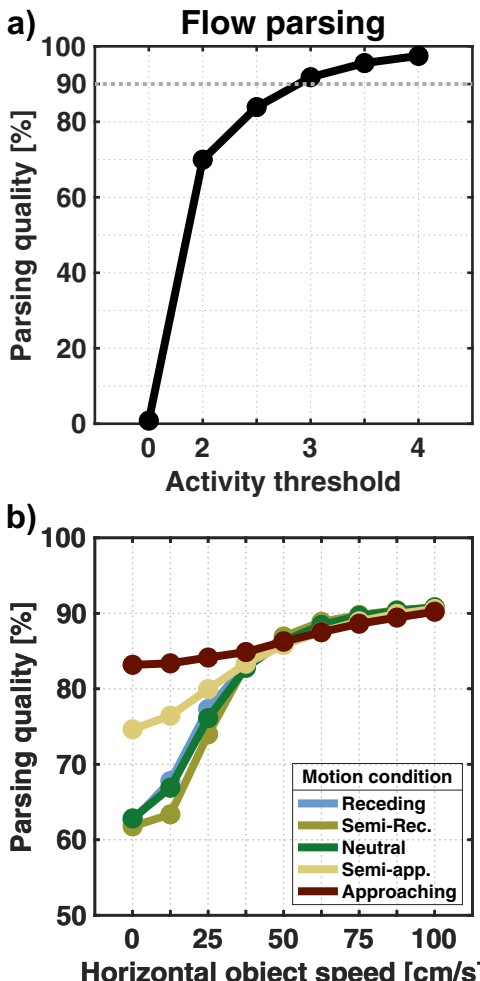

**Fig. 4 | Flow parsing quality. a** Average rate of correctly assigned residual surfaces for different activity thresholds based on simulations without an independent source of motion. Residuals whose activity maximum is below the threshold are used for heading estimation. The dashed line marks 90%, which we aim to surpass by our choice of the threshold. **b** Results for simulation with a moving object. Still, some residuals are only due to observer flow and should be assigned to the heading estimation layer. Lines indicate the average percentage rate at which the flow parsing process correctly parses the residuals into the respective layer.

that the accumulation of residuals with saddle points pronounced enough to be assigned correctly for object estimation results in a surface with an even more distinct saddle point and, therefore, a higher activity maximum, we set $\tau_2 = 1.5 \cdot \tau_1$.

With that, and as expected, the detection rate, as seen in Fig. 5d, shows a similar pattern as the flow parsing quality (Fig. 5a), namely that the faster objects are easier detected than slower ones. The detection rate rises above 97% for the fastest objects in all motion conditions. When there is no horizontal movement, the object is detected semi-reliably only when it approaches the observer, with detection rates of 47.6% and 80.1%, respectively. When the object is neither moving horizontally nor towards the observer, the detection rate is well below 10%. Only 5% of the objects moving such that the combined flow was as fast as or slower than the observer flow while in the same direction were detected. Objects that generated a higher speed ratio were detected more often, reaching detection rates of 50% and 75% at speed increases of 65% and 85%, respectively. These rates were also achieved by objects whose combined flow deviated by 27° and 41° but not in speed.

Preliminary testing showed that increasing the parameter $\tau_2$ does not significantly alter the pattern of the detection rate apart from a reduction in

magnitude. Instead it changes the number of flow fields for which object localization and object direction estimation are performed, as those processes were only performed for detected objects.

Object localization. While object detection is based on the peak value of the activity map computed in the object estimation layer, the object location is determined by the position of the peak. The localization error is the distance between the estimated location and the object's center, if the object is detected. Hence, for objects with a radius of up to 4 dva, estimated locations inside the object can still result in small localization errors. Results are shown in Fig. 5e.

Approaching objects are localized reliably, regardless of horizontal speed. The average localization error was below 2.1 dva in the "semi-approaching" condition and below 0.97 dva in the "approaching" condition. With no horizontal movement, the non-approaching objects were mislocalized with an average localization error between 18 dva and 37.9 dva. This error quickly drops with an increase in horizontal speed, going below 5 dva when the objects move at 25 cm/s and below 1 dva at 62.5 cm/s.

When the combined flow deviated only in speed, the average localization error dropped below 4 dva at a speed increase of 85% due to the object movement. Similarly, a sole deviation in the flow direction of 46° led to an average localization error below 4 dva.

Object direction estimation. The last aspect of the object estimation process is estimating the direction of the object's movement. For this, we considered all objects that were detected and correctly localized in a radius of 10 dva to the object's center. To align with previous research in humans, we restrict our analysis to the "receding" motion condition, where the combined flow was purely horizontal. Results can be seen in Fig. 5f.

As the object in our paradigm is placed relative to the FOE, and this offset placement consists of the direction in which it is placed and its eccentricity, the findings can be easily summarized: Compared to the flow direction, the estimated direction is tilted downwards when the object is above the FOE but tilted upwards for objects below. The estimated direction is not tilted when the IMO is to the left or right of the FOE. Additionally, the closer the object is to the FOE, the smaller the effect. Peaks of the tilt are around 11.9°, 22.4°, and 34.6° in the corresponding direction for objects placed at 5 dva, 10 dva, and 15 dva eccentricity, respectively.

**Flow variations.** Up to this point, the object in our simulations was surrounded by background points, and the flow we used as input for our model was either due to self-motion or its combination with object movement. To test our model's robustness, we deteriorated the flow fields by either introducing directional noise or presenting background flow in only one half of the visual field so that we could include the object in the other half. We focused on the "receding" condition to match existing psychophysical studies that used similar flow field variations. Additionally, we lowered the number of possible object offsets but kept the activity threshold $\tau_1$ the same as before.

Directional noise. For this simulation, flow fields are computed as before. Before presenting them to our model, each background vector is altered by rotating it by a random degree, drawn from a Gaussian distribution, indicating the different noise levels we are testing. These are a "low noise", a "mid noise", and a "high noise" condition characterized by standard deviations of 7.5°, 15° and 30°, respectively. The previous simulation is included as a "no noise" condition.

In terms of flow parsing quality, adding noise lowers the rate at which residuals are correctly assigned (Fig. 7a). With no horizontal movement, the rate drops from 62.5% in the "no noise" condition to around 38% for the three conditions that include noise. However, the improvement in flow parsing quality that comes with increased object speed is still present, with improvements of 21.7%, 14.4%, and 4.5%, depending on the noise level. Generally, the noise raises the overall activity computed from the respective residual surfaces. This leads to fewer residuals ending up in the heading

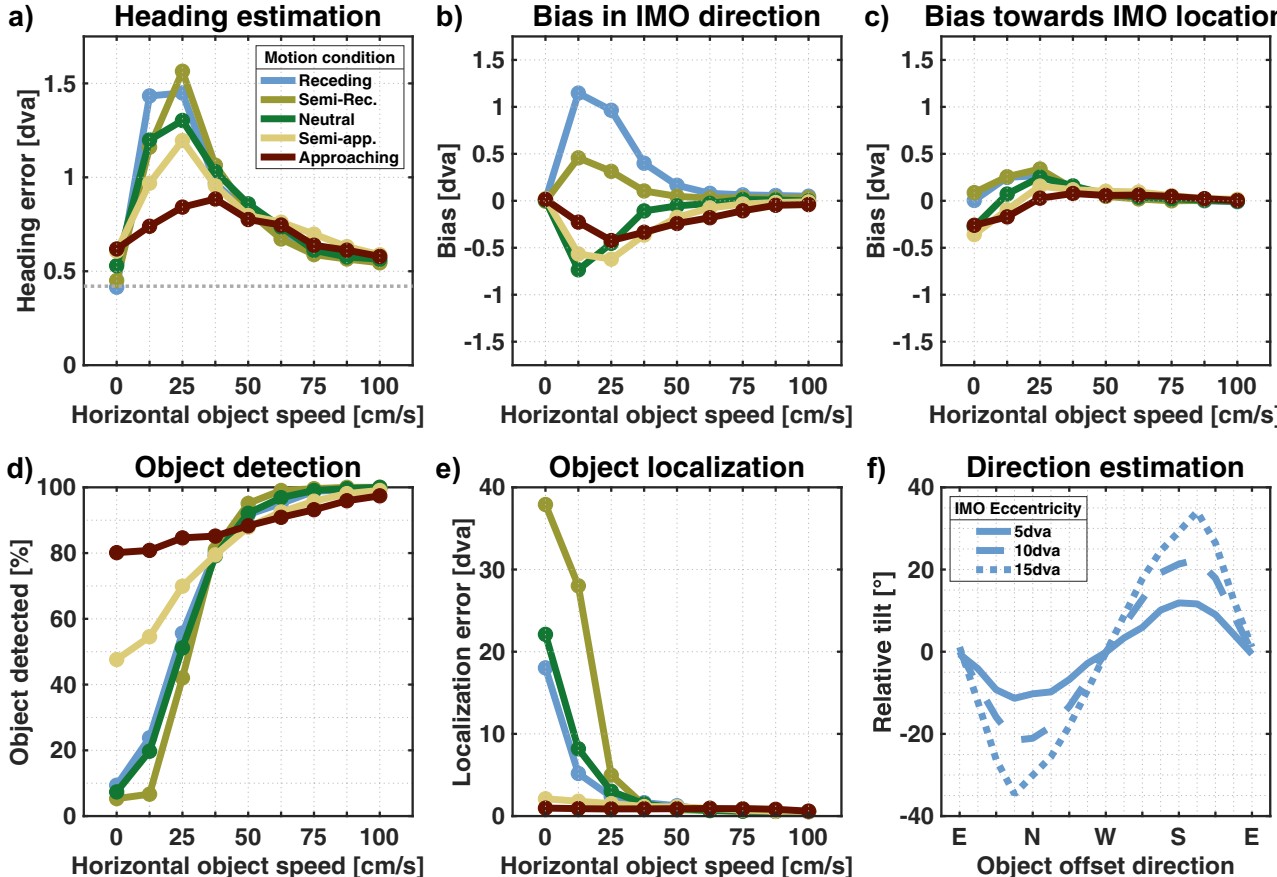

**Fig. 5 | Heading and object estimation results. a** Heading error by horizontal object speed and averaged over all simulations. The dashed line represents the average error for flow fields without objects. **b**, **c** Heading error portion that is in the direction of object movement and position, respectively. **d** Average object detection rate. **e** Localization error for detected objects. **f** Average angle between the estimated object movement direction and the combined flow direction for detected objects moving in the "receding" condition. Results are presented for different object eccentricities, with positive values indicating an upward tilt and negative values indicating a downward tilt.

estimation layer. Without noise, an average of 26.3% of the residuals are used for object estimation, which increases to 62.6%, 71.6%, and 88.1% for the three noise conditions, respectively.

Directional noise leads to an increase in the average heading error to 1.1 dva, 16.65 dva, and 41.98 dva for the respective noise conditions when no object is present and otherwise peaks at 2.63 dva, 22.39 dva, and 44.58 dva (Fig. 7b). Furthermore, only the "low noise" condition retains the previously found pattern (Fig. 5b) in which the heading error increases at first before decreasing for faster object movement.

Regarding the overall pattern of the detection rate, it is kept intact as the rate rises with object speed, no matter the noise level (Fig. 7c). Performance is lower in the "low noise" and "high noise" conditions as the object needs to be faster to reach the same detection rate as in the "no noise" and "mid noise" conditions. Nonetheless, the "low" and "mid" conditions reach a detection rate of 99.2% and 100%, respectively. Even the "high noise" condition surpasses 90% for the fastest objects.

The effect of noise on localization performance is similar to the one on detection rate. A steady improvement in the localization that comes with an increased object speed is present for all noise conditions (Fig. 7d). While the "low noise" and "mid noise" conditions give rise to a localization error comparable to the "no noise" condition, only the fastest objects are reliably localized when flow fields are confounded with the highest noise level.

The object direction estimation based on noisy flow shows the same tilt pattern for the simulation without noise (Fig. 8a). It still holds that the estimated direction of objects placed above the FOE is tilted downwards, and vice versa, while for objects to the left or the right, there is no tilt on

average. The magnitude of the peak tilt is reduced due to the noise, but only slightly for "low noise" and "mid noise" conditions with 19° and 21.6°, compared to the 22.7° in the "no noise" condition. The highest noise level gives rise to the most prominent peak reduction down to 13.1°.

Spatial isolation of the object. In this simulation, background flow was only present in one hemifield, either in the "same" or the "opposite" half to where the object was placed. Observer translation was set to be towards the center of the FOV, and the object was positioned either above, below, or the left or the right of that. The object's size was 1 dva or 4 dva in radius.

In terms of flow parsing quality, despite removing half of the background flow, the overall pattern is still intact as it increases alongside the object's horizontal movement speed (Fig. 7a). The rate of correct assignment of residuals plateaus at a higher level, around 92% and 97% for the "same" and "opposite" conditions, respectively, compared to the base simulation, which reaches up to 90.7%.

The heading estimation is quite similar to the base simulation, no matter which hemifield of flow was removed (Fig. 7b). It peaks at 1.8 dva and 1.2 dva for objects moving at 12.5 cm/s compared to the peak of 1.4 dva at 25 cm/s in the base simulation. For faster objects, the heading error drops to around 0.5 dva in all three conditions.

Object detection and localization performance both show a slight improvement compared to the base simulation, as they already reach the limits of 100% detection rate and a localization error below 1 dva for objects moving 50 cm/s compared to the 75 cm/s needed in the base simulation (Fig. 8c, d).

**Fig. 6 | Performance by flow deviation.** Results for all simulated flow fields are presented by speed ratio and directional deviation. Speed ratio is the quotient of the average speeds of the combined flow and observer flow at the object's location. Values greater than 1 indicate that the combined flow is faster. Directional deviation is the average angle between the respective combined and observer flow vectors. The red dashed lines in the large panels indicate flow fields where the combined flow only deviates in either speed or direction. Running averages along those lines were computed to show the dependence of the performance on either of those aspects. To ensure enough data was available for such a computation, flow fields with combined flow deviating a maximum of 10% in speed and a maximum of 1° in direction, respectively, were included. Results can be seen to the left and below the large panels for the dependence of the performance on directional deviation and speed ratio, respectively. **a** shows the flow parsing quality and **b** shows the heading estimation error. **c** and **d** show the results of the object estimation, the object detection and object localization, respectively.

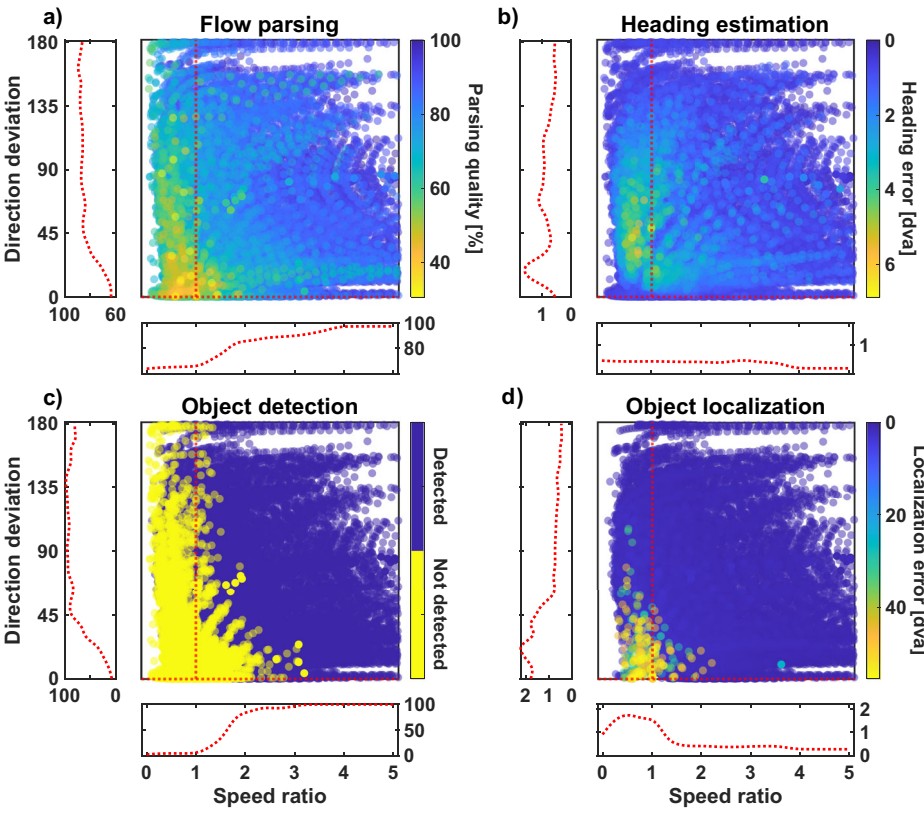

**Fig. 7 | Estimation results for flow variations.** Included are the results for the base simulation (blue lines, "receding" condition), estimations for flow with added directional noise (green lines), and flow fields with background flow only in one hemifield relative to the object location (red lines). **a** Flow parsing quality. **b** Heading estimation error. **c** Object detection rate. **d** Localization error.

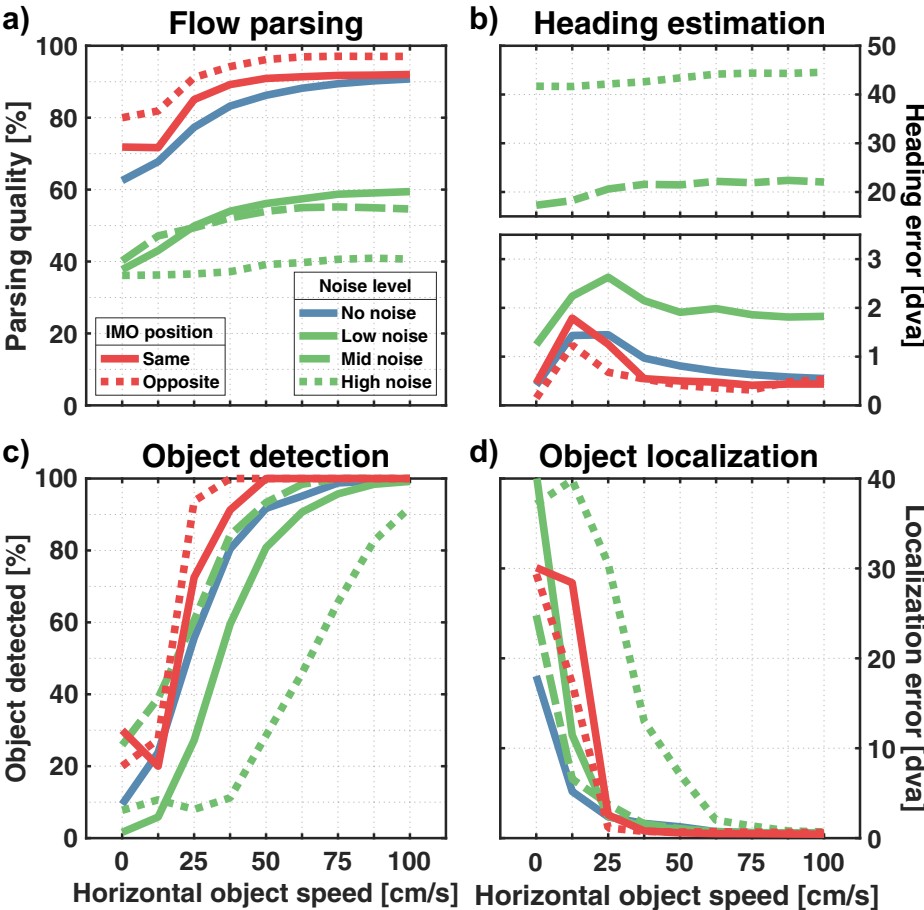

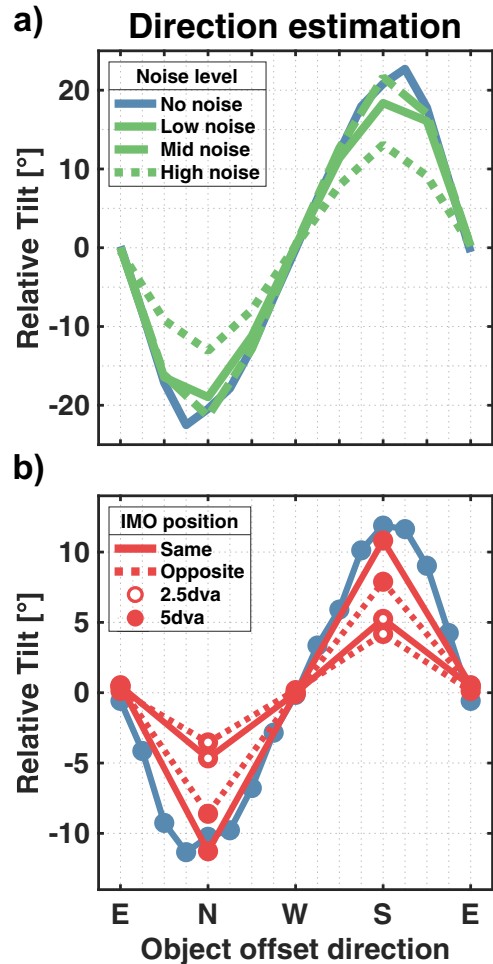

**Fig. 8 | Object direction estimation for flow variations.** Panels show the tilt of the estimated object direction compared to the horizontal combined flow based on the direction of the object offset. **a** Tilt averaged over all object eccentricities and sizes tested for the different noise levels. **b** Tilt results for the spatial isolation simulation. Results were not averaged over object eccentricities as the base simulation did not include objects at an eccentricity of 2.5 dva.

Similar to the previous results, presenting the model with only half the background flow does not break the tilt pattern previously found in the object direction estimations. The estimation remains tilted towards the FOE compared to the combined flow direction with a larger tilt when the object is placed at a higher eccentricity. Overall, the tilt was most prominent when the flow was presented in the whole FOV, peaking at 11.8°. Peak tilt was smaller at 11.3° and 5.2° when the object was placed in the "same" hemifield as the flow at eccentricities of 5 dva and 2.5 dva, respectively. We found an even further decrease in tilt when the object was isolated from background flow with peak tilts of 8.6° and 4.2°.

## Discussion

We presented a computational model to causally separate sources of motion in optic flow fields and recover information about self-motion and an independently moving object. The optic flow we used is represented by a vector flow field derived from a simulated self-motion scenario in which the object's size, position, and velocity are systematically varied. Given an optic flow field as input, our model efficiently estimates the direction of self-motion, detects and locates an independent source of motion, and estimates the direction of its movement. Part of our model's flow field evaluation process is to compute residual surfaces for flow from different parts of the visual field that indicate the consistency of the flow pattern with self-motion. Depending on the level of consistency, the surfaces are used either for

heading or object estimation. While most of these aspects were already the subject of research over the years, studies often only covered one. To show that our model is capable of reproducing many of the reported behavioral patterns simultaneously, we developed a coherent simulation paradigm instead of recreating the stimuli used in all the different experiments.

### Model performance and comparison to human studies

In basic self-motion estimation tasks, humans are known to be able to determine their heading from optic flow within 1–2 degrees of error when the flow is the result of simulated movement through a rigid scene[3,4,14]. Our simulations align with that, as the average heading error for flow fields without moving objects is 0.42 dva.

The simulations of the FLOW PARSE model revealed that a systematic variation of the object speed produces a particular pattern regarding the magnitude of the heading mis-estimation. These results match the findings of Dokka et al., who described that the dependence of heading error on object speed showed small mis-estimations for the slowest and fastest objects and peak error for intermediate speeds[30].

Furthermore, the simulations revealed that the direction in which heading estimation is biased depends on the object's movement direction and whether there is relative motion in depth between the object and the observer. These dependencies were found in several studies in human psychophysics[25–28,60]. However, similar to our simulation, no such bias was seen towards the location of the object[28]. The original population heading map model explained the bias in the object movement direction as the most consistent explanation for the flow field with purely lateral combined flow to be a combination of a shifted observer translation and a rotational component to counteract the shift for the rest of the flow field[28]. On the other hand, the explanation for an approaching object was a translation with an offset angle. Our heading estimation is also based on heading maps computed with the subspace algorithm, so our explanation for these findings is the same.

Our model reliably detected objects based on deviations of either speed or direction. Similarly, humans are able to detect an independently moving object whose only cue was a disruption of a self-movement flow pattern, either in terms of a change in direction[6] or a change in flow speed[7]. They found detection rates of 75% for directional deviations of around 15° and speed increases of around 40%.

These object detection studies only asked their participants to indicate if they noticed a self-moving object, but not to localize it. We are unaware of other studies that included a localization task for objects in optic flow fields, so we cannot compare our model's performance to behavioral data. Instead, our model would predict that the localization of detected objects based on the saddle point in a residual surface should work well.

In general, in our simulations the differences between motion conditions start to dwindle for objects moving sufficiently fast horizontally (>25 cm/s). This holds for the heading and localization error and the detection rate, though not for the directions in which self-motion estimation is biased. A critical movement speed is also present for the flow parsing quality. The difference between motion conditions here depends solely on the residuals due to combined flow and whether they are correctly assigned to the object estimation layer. Combined flow when the object approaches the observer while barely moving horizontally tends to be faster than when only the observer is moving. Therefore, the flow parsing quality for those conditions is higher than that of the remaining ones. When the object moves fast enough horizontally, the speed ratio or the direction deviation is sufficient so that residuals due to enough object motion are reliably used for the object estimation. The surfaces that remain wrongly used for heading estimation only carry small amounts of information about the object movement, which are those for which the corresponding operator receptive fields only contain parts of the object. Hence, those residuals perturb the heading estimation only slightly, although distinguishable in the bias direction. On the other hand, objects that move that fast are already reliably detected and localized, as most of the corresponding residuals are used for the object

estimation, regardless of the motion condition. If human data is acquired with a similar multi-task paradigm, it would be interesting to see if such a critical speed manifests, allowing for reliable causal attribution and, therefore, performance alignment across the motion conditions.

When participants were tasked with reporting an object's trajectory in a flow field, the indicated direction was tilted towards the FOE compared to the on-screen motion, and the effect's magnitude increased for objects farther away from the FOE[8–11]. These responses were consistent with a scene-relative object motion. If the on-screen movement were interpreted as a result of self- and object motion, the tilt direction and its change in magnitude would match the subtraction of the flow component due to self-motion. As our model shows similar patterns for that task, we conclude that the direction that our model estimates is the object's world-relative movement direction in retinal coordinates rather than the direction of the combined flow.

Adding directional noise to a flow field affects the heading estimation process in our model twofold. First, even pure self-motion flow has reduced reliability as a cue for heading. Second, due to the increased activity, fewer residual surfaces end up in the heading estimation layer. While this leads to an increase in the heading estimation error, the previously found pattern in which the heading error increases at first before decreasing for faster object movement is not lost for the "low noise" condition.

Contrary to the heading estimation layer, the object estimation layer receives a larger share of the residuals due to the noise, even though the number of residuals with object information does not change. Therefore, the object estimation results might be diluted due to wrongly assigned residuals. Regarding object detection, the model can still detect an object if it moves fast enough, regardless of the noise level. However, the need for a higher speed in the "low noise" and "high noise" conditions to detect the object indicates that the noise-induced increase in activity for the individual residual surfaces does not directly translate to higher activity for the summed-up residual surface. Additionally, the assessment of scene-relative object movement was largely independent of noise level. These results are broadly consistent with several human studies that showed that self-motion estimation works reasonably well even when flow fields are confounded by various types of noise[23,24,61]. The subspace algorithm, which forms a key part of our model, was also shown to be relatively robust when presented with noisy flow fields[39]. Foulkes et al. showed that the algorithm is capable of producing similar patterns of estimated heading thresholds as humans, albeit with considerably higher average error[61]. Regarding object direction estimation, Foulkes showed that, compared to noiseless flow fields, the introduction of noise decreased the peaks of the relative tilt pattern[62].

When Warren and Rushton removed flow from one hemifield to isolate the independently moving object from the background flow, their motivation was to rule out that the assessment of scene-relative object motion was due to local motion interaction[11]. As the participants' reports of perceived object trajectory revealed the previously found tilt pattern, they concluded that this process is global. Our model aligns with this assessment, as it also retains the tilt pattern, including the slight reduction in the "opposite" condition that the participants showed. Furthermore, the only local motion process implemented is the vector averaging taking place in layer 1. Neither the computation of residual surfaces nor any later process relies on local motion. Some types of models rely on computing local flow differences to estimate heading, as this removes a potential rotational flow component[49–52]. In contrast, the subspace algorithm can, by design, solve flow fields for the translation direction even in the presence of rotations. Overall, the model results match those reported by Warren and Rushton[11].

## Further outlook

Over the years, many studies have approached various aspects of flow processing, and the variety of paradigms used is appropriately large. Our goal was to design a paradigm that allows the simultaneous estimation of self-motion and object parameters to cover various findings. By design, we neglected some of this paradigm variety. We have not covered whether our model could deal with environments different from the generic cloud of points we used or scenes that encompass rotational components. For example, a ground plane is another environment often used in optic flow studies[3,21,22,24] and simulations of the original heading map model[38]. We expect this to pose no significant challenge to the FLOW PARSE model. While the reduced amount of flow due to the lack of points in the upper hemisphere would give rise to fewer residual surfaces to process, the available depth structure should be sufficient for our flow processing to work. However, a translation towards a wall, a scene lacking depth, gives rise to ambiguous flow fields that can be explained with different heading directions[2,15,56]. As the subspace algorithm has shown this ambiguity resulting from residual surfaces with two distinct peaks, this might pose a challenge for the saddle-point operators employed in the third layer located between those peaks. However, this problem could be mitigated by the first flow processing step, as the vector averaging in the first layer might introduce enough noise to reduce the ambiguity.

Regarding conditions that involve rotational observer movement in addition to translation, while the flow fields would grow in complexity due to the added rotational component, the heading estimation based on the subspace algorithm is designed to be rotation invariant. This invariance should expand to our flow parsing process and make our model robust against rotations. Since there is no human data on such conditions yet it would be interesting to explore both model and human performance in the future.

## Flow parsing and causal inference

The flow parsing hypothesis proposed that the brain uses its sensitivity to self-induced flow patterns to isolate object flow, following evidence that humans can judge the scene-relative motion of an object from retinal motion alone[8–11]. While this process was not further specified, its description was often accompanied by the illustrative idea of "subtracting" a flow field due to self-motion from the retinal flow, because, despite the lack of plausibility for such an explicit subtraction process in the brain, it conceptually allows for easily understood predictions and reports. This has led to the thought that heading estimation is a prerequisite of flow parsing, since the flow component due to self-motion would need to be identified first, before it can be subtracted from the flow to reveal object motion. Rushton and colleagues have since clarified that this is not the case, and that flow parsing may not rely on prior heading estimation[34,35].

The flow parsing mechanism implemented in our model differs from the above approaches. But before pointing out the differences, we want to highlight the similarities. Most importantly, our implementation is a process that, among other things, assesses the scene-relative motion of an object while relying solely on visual input. In addition, it relies on sensitivity to flow patterns due to self-motion in the form of heading likelihood maps, which have already been used to model population responses of MST neurons[37,41].

Our model differs from the original proposal in that we do not aim to isolate non-observer flow. Instead of working at the level of retinal flow, we propose a mid-level mechanism working on the population heading map. Our process infers the causal sources of the likelihood maps and, based on the result, passes them directly to the corresponding processes. The lack of a need to loop back to the retinal flow level for further processing may be a significant increase in processing speed. Furthermore, in our model heading estimation is not a prerequisite for flow parsing, but rather one of its results. This allows us to find the particular speed dependence of the heading error reported by Dokka et al.[30]. When the object moves slowly, the combined flow is mistakenly attributed to self-motion alone, even though it is not a valid cue. On the other hand, for faster objects, the flow is reliably recognized as being due to independent motion and used accordingly, reducing the impact on heading estimation performance.

Placing the flow parsing process at that mid-level stage of flow processing matches better to the results of studies in the recent years. Heading estimation is neither a prerequisite[34] nor limits the ability to assess scene-relative object motion[35]. While directional noise directly affects the flow parsing quality by increasing the overall activity in layer 3, which impacts

both types of processes, it additionally deteriorates the heading estimation performance. The reduced flow quality due to high levels of noise can render self-motion estimation ineffective but only lessens the tilt found in object direction estimation, leaving the overall pattern intact.

By separating the sources of retinal motion our flow parsing model can be seen as a process of causal inference. Traditionally, causal inference is modelled as a Bayesian integration process in which, for example, two sources of motion and their variances are optimally combined (e.g., ref. 30). While clearly many estimation processes in the visual system are Bayesian-optimal in this sense, such a view on causal inference does not address the question how heading and object motion are derived from the patterns of retinal motion. The Bayesian inference process initially requires knowledge of the true heading and object motion along with their measured variance. Our model differs from this approach by focusing on the computational mechanisms by which heading and object motion can be estimated directly from the retinal input. It is astounding that the results of this computation match the results of the Bayesian causal inference account[30]. It indicates that the computations of the model implement an optimal procedure for identifying separate sources of retinal motion.

## Conclusion

This detailed discussion of experimental results and the prior introduction to various types of models aimed to show the unique position in which our model fits. While it provides different types of estimations, ranging from self-motion to object estimation, it reproduces behavioral trends found in various studies. The key to that is the flow parsing process we implemented, which is based on sensitivity to optic flow patterns consistent with self-motion. Our interpretation of the flow parsing process works on the residual surfaces computed for parts of the optic flow and does not parse the flow itself into its components. While this could be achieved retrospectively, it is not necessary in the context of our model, as estimations of all the parameters are based on these surfaces and not isolated flow. Additionally, the straightforward structure of the model shows that recurrent connections or feedback loops might not be necessary for those processes. In our model, heading estimation and independent object motion estimation work in parallel and in a fast, feedforward fashion.

## Methods
### Mathematical descriptions of optic flow

Mathematically, optic flow arising from self-motion in a static scene is described by a set of flow vectors. These vectors represent the image velocity, that is, the time derivative of the spatial components of the projection of the 3D points in the environment onto the image plane. This plane is placed at distance $f$, perpendicular to the line of sight, and acts as a simplified representation of the retina of the monocular observer. The relationship between the instantaneous observer movement and the optic flow field can be described with the following equation:

$$v(p) = \frac{1}{Z} A(p) T + B(p) \Omega,$$

that describes the image velocity for a static scene point $P = (X, Y, Z)$, with $p = (x, y) = f \cdot (X/Z, Y/Z)$ as the image coordinates, and $T$ and $\Omega$ as the motion parameter of the observer, the translation direction and rotational velocity, respectively[2,39].

$$A(p) = \begin{pmatrix} -f & 0 & x \\ 0 & -f & -y \end{pmatrix},$$

$$B(p) = \begin{pmatrix} xy/f & -f - x^2/f & y \\ f + y^2/f & -xy/f & -x \end{pmatrix}.$$

Written in this form, this equation illustrates that optic flow fields consist of two components, where only the translational part, not the

rotational part, depends on the depth. If P belongs to a non-static scene object with $S$ describing its translation direction in the world, the equation can still be used to compute the respective flow. For $\Omega = (0, 0, 0)$ it holds:

$$\begin{aligned} v_c(p) &= \frac{1}{Z} A(p)(T - S) + B(p)\Omega \\ &= \frac{1}{Z} A(p) T + \frac{1}{Z} A(p)(-S) \\ &= v(p) + v_O(p), \end{aligned}$$

describing the combined flow due to the relative motion (T-S) between the observer and the object, with $v_O$ as the object flow.

### Subspace algorithm

Trying to find the best self-motion parameter to explain a given flow field imposes the challenge of solving a set of equations with many unknowns: the 6 self-motion components and depth values of every point in the scene that contribute to the flow field. While there are different approaches to solving this problem, this work focuses on a particular method, the subspace algorithm developed by Heeger and Jepson[39]. This method allows for successive solving for the observer translation direction, then the rotation, and finally the depth structure of the scene based on the prior results. The relevant part for this study is the first one. To solve for the translation, a type of residual function is defined that calculates how consistent a candidate translation direction is with the given flow vectors, allowing for a least-square estimation approach by picking the candidate direction that minimizes the residual value. To define said residual function for a candidate direction $T$ and given flow $v$ at image locations $p_1 = (x_1, y_1), \ldots, p_n = (x_n, y_n)$ we define

$$C(T) = \begin{pmatrix} A(p_1)T & \cdots & 0 & B(p_1) \\ \vdots & \ddots & \vdots & \vdots \\ 0 & \cdots & A(p_n)T & B(p_n) \end{pmatrix}$$

with the matrices as defined above. The residual function is now defined as

$$R_v(T) = \parallel v^t C^\perp(T) \parallel^2,$$

with $C^\perp(T)$ depicting the orthogonal complement to $C(T)$. Sampling the set of all candidate translation directions, the heading space, in a retinotopically organized way and evaluating the corresponding residual functions on it yields a surface of residual values that maintains the same retinotopic organization. Such a surface can be seen as a heading likelihood map.

### Model simulations

**Paradigm.** In the literature on self-motion and flow parsing a variety of paradigms have been employed to study various aspects of human perception, each well-designed to isolate a particular function or parameter set. Each of these paradigms would require a different set of settings and parameters for a model simulation (e.g., location, size, speed and direction of the object, speed of observer motion, size of the FOV, dot density, etc.). As we aim to show that our model is capable to simultaneously reproduce different facets of human behavior we decided to avoid implementing the individual specifics of each study and rather develop a generic simulation paradigm that allowed to compare the model behavior to essential findings across studies.

That paradigm establishes a self-motion scenario that consists of simulated observer translation towards a static cloud of dots. The cloud starts at 4m from the observer and has a depth of 6m. Dots in the cloud are placed so they are randomly distributed in the viewing window with height and width of 70 dva with 0.55 dots/dva². Additionally, the scene contains an opaque object consisting of 50 dots, regardless of its size. These object dots are either all 4m from the observer or randomly placed inside the depth range the cloud covers. The object takes the form of a

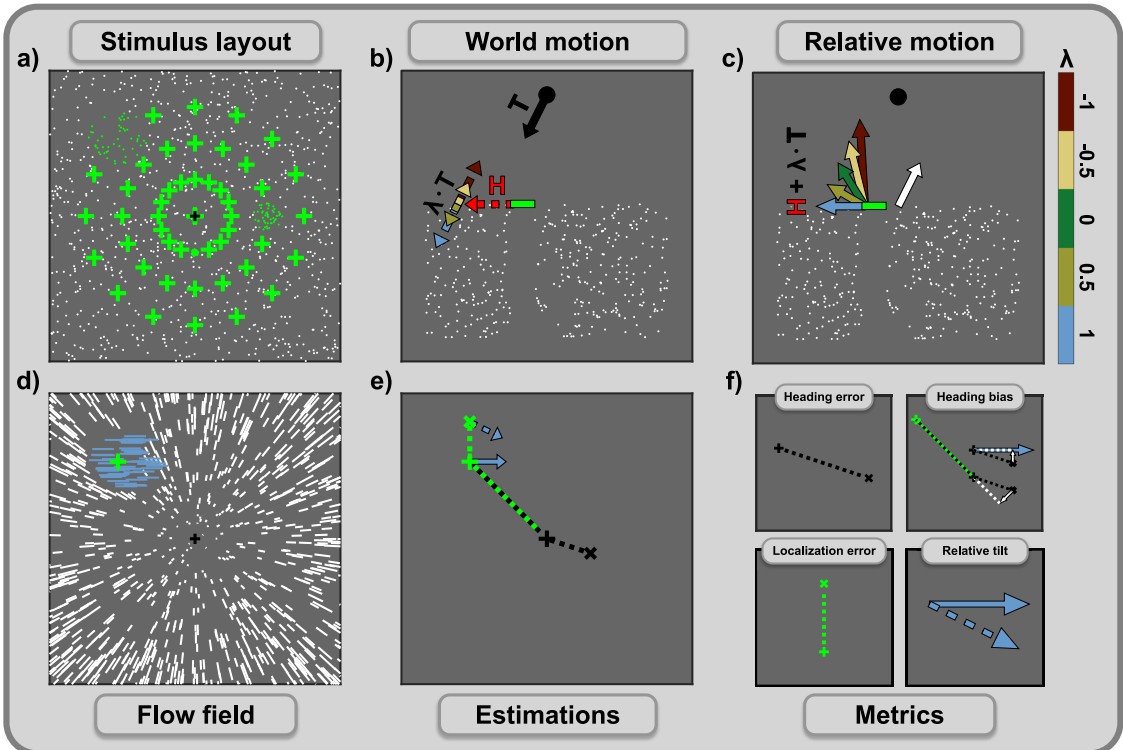

**Fig. 9 | Simulation paradigm, estimation results, and performance metrics. a** The object's location in the visual field is determined relative to the direction of the observer's movement (black plus). Potential object positions vary evenly in eccentricity and placement direction. While the simulation contains scenes with only one object present, the panel includes examples of three objects that vary in size and location. Remaining potential object positions are indicated with a green plus. **b** The scene consists of the observer's motion $T$ (black arrow) toward a cloud of stationary dots (white dots), presented in a top-down perspective. The self-moving object (green bar) is opaque and occludes parts of the scene. Object movement consists of two components: a horizontal movement $H$ (red dashed arrow) at different speeds and a movement along the direction of the observer's translation ($\lambda \cdot T$, colored dashed arrows). This component defines the motion condition and ranges between the "receding" (blue) and the "approaching" (dark red) condition. **c** The scene in an observer-centered coordinate frame, again in a top-down perspective. Observer and object motion is combined and converted to relative motion between the points of the scene and the observer to compute the flow fields. **d** Flow fields are calculated for each combination of object size and location, the motion condition, and horizontal object speed. The example flow field shows the result of a simulated scene in the "receding" condition, in retinal coordinates. Combined flow is horizontal due to the backward motion canceling any changes in depth between the object and the observer. The black plus and the green plus indicate the observer's translation direction and the object's location, respectively. **e** The model estimates various scene parameters and whether the object is present for each flow field. The estimations of the heading direction (black x), the object location (green x), and the object's movement direction (dashed blue arrow) are shown along the actual parameters, in a retinal coordinate frame. The green-black dashed line indicates the object offset relative to the heading direction. **f** Different metrics to measure model performance based on the estimations shown in (**e**). Heading and localization error represent the distance between estimation and the true parameter in dva. Potential heading biases are indicated by the projection (solid white arrows) of the mis-estimation vector (black dashed line) onto the object direction and the offset vector. Here, the estimation was biased in object direction, as the projection is on the direction vector, and opposite of the object's location, as the projection lands on the backwards extension of the offset vector. The estimated object direction is measured relative to the object flow direction by calculating the angle between them.

circular shape with a diameter of 1 dva, 4 dva, or 8 dva and is placed relative to the translation direction of the observer. This offset is determined by its eccentricity to the FOE, ranging from 0 dva to 15 dva, in steps of 5 dva, and one of 16 directions, starting from the one to the right in steps of 22.5° counter-clockwise.

Observer motion direction was randomly picked from the innermost 10 dva of the FOV, the corresponding translation in the world $T$ calculated, and the speed set to 2 m/s. Object motion in the world is defined as

$$H + \lambda \cdot T,$$

with the first component $H$ as a horizontal translation to the observer's right, ranging from 0 m/s to 1 m/s, in steps of 0.125 m/s. The second component, depending on the observer's translation direction $T$, falls into one of three categories: (a) the object moves in the same direction as $T$ so that it recedes from the observer ($\lambda > 0$), (b) by moving in the direction -$T$ the object actively approaches the observer ($\lambda < 0$) and (c) the second component is absent, making the object's movement independent from the observer's translation ($\lambda = 0$). With $\lambda$ ranging between 1 and −1, in steps of 0.5, this

setup includes the two motion conditions used typically in similar studies[25–28] to be the edge cases of our continuous set of motion conditions (see Fig. 9). We will refer to the motion conditions with the corresponding values of $\lambda$ or as "receding", "semi-receding", "neutral", "semi-approaching", and "approaching", as they describe the object's movement in depth. For example, with $\lambda = 1$, the object is "receding" in the world, as its depth movement component is identical to the observer's. Moving in the opposite direction with $\lambda = -1$ makes the object "approaching". With $\lambda = 0$, so without an in-depth motion, the object movement is "neutral". Similar to the study of Li et al.[28], all object dots were placed at the front of the cloud for the "receding" condition but varied in depth otherwise. Combinations of all the possible object offsets and sizes give rise to 147 different spatial layouts. We created five self-motion scenarios for each layout, then simulated observer motion and all object movements mentioned above for a total of 45 flow fields per scenario. Each flow field is then used as an input for the computational model.

**Model parameters.** The heading space, the set of all candidate heading directions for which residual values are computed, covers the central

86 dva × 86 dva of the visual field. Candidates are sampled on a hexagonal grid so that adjacent directions are 1 dva apart.

The model parameters were set to the following: First layer operators are placed on an evenly spaced rectangular grid so that the minimum distance to candidate directions is 0.5 dva. Avoiding placing candidates and vectors of the flow representation too close reduces the possibility of irregular results during residual computation[63]. Receptive fields of the first layer operators have a radius of 2 dva. The receptive fields of the second layer operators have a radius of 20 dva, are spaced 12 dva apart and evenly placed to cover the FOV. This results in 2760 vectors in the optic flow field that is the output of the first layer and 36 residual surfaces calculated, one for each of the operators in the second layer. The groups of operators in the third layer that are placed on the residual surfaces are spaced 1 dva apart. Sizes of the corresponding circular areas range from 1 dva to 5 dva in radius and the orientation of the cross-shape arrangement from 0° to 75°, relative to the cardinal axis. Activity threshold $\tau_1$ ranged between 0 and 4 and the detection threshold was set to $\tau_2 = 1.5 \cdot \tau_1$.

**Performance metrics**. Certain metrics must be introduced to assess the model's performance. A fundamental part of this model is the flow parsing that assigns residual surfaces to one of the subsequent layers. Measuring the quality of that assignment is crucial to determine the model's capabilities because both processes, heading and object estimation, depend on the input to the corresponding layers. We evaluate *flow parsing quality* by calculating the rate at which our activity map criterion agrees with the actual causal source.

The following metrics examine the quality of the parameters that the model estimates. These metrics are visually depicted in Fig. 9f. The primary metric to rate the heading estimation is the *heading error*, which is the distance between the true translation direction and the estimated translation direction in dva. Additionally, we calculate the extent to which the heading estimation error is in the direction of the object's movement direction or its location, to unveil a potential *heading bias*. This is done by projecting the heading error vector onto the object direction and the offset vector, respectively.

Lastly, there are the metrics for the object estimation. We consider three aspects when measuring the object estimation performance of the model. For *object detection*, we check whether the binary estimation regarding the object's presence correctly reflects the stimulus's state. Similar to the heading estimation error, *localization error* is calculated as the distance between the estimated location and the retinal position of the object's center. The estimated direction of the object's movement is in retinal coordinates. We compare the actual direction of the combined flow with the estimated direction by calculating the angle between the corresponding vectors, resulting in the *relative tilt*. The sign of this angle reflects the direction in which the estimation is tilted, with positive values indicating a tilt in counterclockwise direction and negative values a tilt in clockwise direction.

## Reporting summary

Further information on research design is available in the Nature Portfolio Reporting Summary linked to this article.

## Data availability

The data that support the findings of this study is available at the Open Science Framework repository with the identifier https://doi.org/10.17605/OSF.IO/CZ4E6[64].

## Code availability

The MATLAB code (The Mathworks Inc, Natick, MA, USA)[65] used in this study is available at the Open Science Framework repository with the identifier https://doi.org/10.17605/OSF.IO/CZ4E6[64].

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

## Acknowledgements

This work was supported by the German Research Foundation (DFG La 952-10) and has received funding from the European Union's research and innovation programme under the grant agreements No 951910 (MAIA) and 101086206 (PLACES).

## Author contributions

M.S. and M.L. designed the model and the simulation paradigms. M.S. conducted simulations, analysed the data, and wrote the original draft of the manuscript. M.S. and M.L. finalized the manuscript.

## Funding

## Competing interests

The authors declare no competing interests.
