## [Transparent Peer Review file · Communications Biology]

Flow parsing as causal source separation allows fast and parallel object and self-motion estimation

Corresponding Author: Mr Malte Scherff

This manuscript has been previously submitted at another journal. This document only contains information relating to versions considered at Communications Biology.

Version 0:

Reviewer comments:

Reviewer #1

(Remarks to the Author)

Brief summary:

The first wave of research on optic flow focused on the 'rotation problem', how the visual system decomposes the flow field into components for observer translation (heading) and observer rotation. The most recent wave has focused on the problem of 'flow parsing', how the visual system separates the flow field into a global component due to self-motion (heading) and a local component due to object motion. Such flow patterns have been shown to produce specific human biases in both perceived heading and perceived object motion. In the present manuscript, the authors show that the leading model of the first problem (Heeger & Jepson, 1992; Lappe & Rauschecker, 1993) generalizes to the second problem – and naturally reproduces the observed pattern of human biases.

Overall impression:

The manuscript is a tour-de-force of modeling that succinctly explains a variety of human data and offers a coherent solution to two major problems of optic flow processing. First, the authors review the literature on flow parsing and summarize the puzzling pattern of biases that have been reported. Second, they extend their earlier neural model by cleverly splitting the output into a heading map and a moving object map. In a systematic set of simulations, they show that the model, with no further modification, explains the variety of human biases in both perceived heading and perceived object motion. The model provides a computationally unified and theoretically satisfying explanation of the flow parsing problem and the rotation problem, showing that they have a common root in flow field information. To top it off, the solution corresponds to Bayesian-optimal causal inference (Dokka, et al., 2019) without assuming Bayesian priors. This is an impressive piece of work that significantly advances our understanding of a central topic in human and machine vision.

Specific comments:

I only have a few comments, which can be addressed in a minor revision:

1. Please clarify the parameterization of object motion in the self-motion scenario (lines 530-552). I take it that observer translation toward the cloud is a vector with a magnitude $T = 2$ m/s in world coordinates. If the object is moving at $1 \cdot T$ "in the world," then it would move in the same direction and at the same speed (2 m/s) as the observer, with no relative motion ('neutral'). Yet the authors say that $1 \cdot T$ corresponds to the object 'receding' from the observer, whereas $0 \cdot T$ is 'neutral'. What was the coordinate frame for object motion?
2. A crucial aspect of the model is described very quickly and needs to be explained more clearly. Specifically, to what do the different residual surfaces in Layer 2 (Fig. 2, top) correspond? Based on Lines 401-411, I take it that each residual map represents the consistency of candidate headings all over the visual field with the output of a local operator for one receptive field. Is that correct? Please explain in the text.
3. The flow parsing results strongly depend on the value of the activity threshold τ_1 for detecting a saddle point in Layer 3. Lines 648-668 say that τ_1 was selected so that a pure self-motion flow field was assigned to the heading estimation layer 90% of the time. Similarly, the detection of a moving object strongly depends on a second activity threshold τ_2 for a saddle

point in the object estimation layer (call it Layer 4b). Lines 830-837 say that $\tau_2 = 1.5 \cdot \tau_1$. Please clarify why these values (90% and 1.5) were chosen. Given that there is no independent estimate of the values, would it be possible to fit them by reproducing the pattern of a subset of human data, and then to predict remaining human data, in some type of cross-validation?

4. P. 13-17: The Results section begins to bog down in detail about here. The authors might be more selective about what they describe in the text to focus on the main points, and rely on figures more to express the details.

Details:

- Line 303-307: This sentence could be clarified: "What the previous implementations have in common is that only the peak of the computed surface was taken as the final heading estimate, with no further attention paid to the distribution of residual values."
- Line 324 ff: The description of the residual surfaces in Figure 1 is a bit vague and difficult to understand, so it could be made more precise.
- Line 603: Should refer to Fig. 3F.
- Figure 3: Fig. 3B and 3C are not particularly helpful because they are hard to interpret. Is this a top-down view of observer translation (red arrow) toward a cloud (white dots)? The cloud looks like a plane of dots receding in perspective, rather than a cloud in the viewing frustum. The streak of green dots looks like it lies on the plane, not like an object moving through (or occluding?) the cloud. I'd suggest that drawing a square cloud of dots (rather than a trapezoid) would help. Fig. 3E and 3F are impossible to interpret because the viewpoint is not clear – does it represent a frontal view like Fig. 3A and 3D, or a top-down view like Fig. 3B and 3C?

Reviewer #2

(Remarks to the Author)

In their paper "Flow parsing as causal source separation: A computational model for concurrent retrieval of object and self-motion information from optic flow", the authors generate just that; a model that allows for estimation of self-movement and object movement from optic flow concurrently. This is a mechanism that has been previously reported after behavioural investigation, but no model has yet, to my knowledge, modelled this behaviour. This modelling appears to closely match behaviour in heading and scene-relative object movement tasks, without assuming that one is necessary for the other (another recent finding in the field). This model provides the underpinning mechanism that may be at play when we perform flow parsing which is an intriguing thought. The work is convincing, though I feel that a more convincing story could be told, as will be mentioned in my comments below. Perhaps some data showing that this model fits human data better than other previous models would strengthen the conclusions of the paper, but these are clearly, and with reason, beyond the scope of this paper and I would not be surprised if the authors are already looking into that for a subsequent publication. This work is a computational model, so it is infinitely repeatable with the provided codes, I thank the authors for that.

As mentioned above, I feel that there are structural changes that would help to give the reader a better understanding of the gap in the research that this model fills, while making the results section more digestible and allowing the discussion section to contain a more glowing review of the model. Alongside this, I have some points of intrigue that I would appreciate the authors' thoughts on. Please find more detailed comments about these below.

I should also point out that there were a few situations where the grammar could be improved.

Detailed Comments – Structure:

When reading section 5.2 "Comparison to other computational models", I felt that there was little in the way of discussion or comparison between other models and your model (until the paragraph starting on line 1350). Instead, this felt like an introduction to the other computational models that exist and a well written unpacking of the gap in the research that your model fills. For this reason, I suggest that this section could be rebranded and used as part of the Introduction, in a section outlining current computational models of heading estimation and IMO movement estimation and identifying that, previous to yours, there is not a model that sufficiently explains heading estimation and IMO movement estimation concurrently.

This will leave a gap in the discussion section, which, I think can be filled by simplifying the results section. I feel that comparing the model to human performance (lines 738-743), making conclusions about the results (lines 809-820) etc. should form the basis of section 5.1 of the discussion section. Each subsection of the results section can be linked to in the discussion section e.g. "in section 4.3.1, our model was able to detect moving objects with a detection rate of x% at speed increases of y% and deviations of zdeg. This result fits with studies with human participants who showed...". I think that this change will make the results section shorter and more direct with its presentation of what the model outputs, while allowing for clear description of how the model fits with human data in the discussion section.

I think that the discussion section containing a short recap of each result and then linking that directly to other research would provide a more glowing review of the model and what it achieves.

In Results section 4.1, the first two paragraphs discuss the selection of the activity threshold T1 based on the percentage of residual surfaces that are assigned to the self-movement detection layer of the model when there is no IMO present. As this is "the first step in testing the model", I feel that it should have its own section, perhaps section 4.1.1 "Selecting T1". I would then denote the next section as section 4.1.2 "Speed Ratio", or something similar.

Detailed Comments – points of intrigue:

The third paragraph of section 4.1 discusses how changing the horizontal object speed affects the percentage of correct assignment, shown in Figure 4B. I am intrigued as to why the receding, neutral and semi-receding conditions produce similar percentage of correct assignment, while there is a big difference between those conditions and the approaching conditions. Is this because of how these conditions affect the speed ratio?

In fact, I feel that your description of the speed ratio could potentially benefit from a figure which explains what is meant by observer flow, object flow and combined flow, as I often found myself trying to imagine what each of these would be like in

different scenarios.

It also intrigues me that the object speed value whereby the parsing quality is effectively equivalent (37.5cm/s in Figure 4B) across the conditions, lines up quite nicely with the object speed value whereby the heading estimation, bias towards IMO location, object detection and object localisation measures (in Figure 5A and C-E) begin to be somewhat similar. Bias in IMO direction also appears to be similar above this object speed, though inverted for the receding conditions. I don't think this was mentioned in the text, but it could be an interesting finding if replicable with human participants (a critical speed value for consistent object motion perception with approaching/receding objects?)

In section 4.2.2 you simulate isolated IMOs which create largely similar results to when those IMOs aren't isolated. This makes sense as your model uses the flow across the whole receptive field for its processing. As I understand it, the reason behind isolating the IMO in Warren and Rushton (2009) was to remove the possibility of local motion processing (effectively processing the differential of the flow, rather than the flow itself). Could a version of your model be conceived where local motion processing was possible? How do you think this would change your results?

Dr Joshua Haynes

Reviewer #3

(Remarks to the Author)

Scherff & Lappe present a paper entitled "Flow parsing as causal source of separation: A computational model for concurrent retrieval of object and self-motion information from optic flow". This paper describes a new model that simultaneously solves two major problems of motion processing (heading estimation, and flow parsing) in the context of perception and action.

I genuinely enjoyed reading this paper. The authors have done a great job of summarizing and synthesizing a large (and sometimes confusing) literature while also generating an interesting and novel computational approach for solving an important set of perceptual problems.

My major concerns are as follows:

A. One of the strengths of this paper is the clear manner in which it reviews and synthesizes the flow parsing literature. I think this is a great thing to do, particularly as the flow parsing literature can be quite confusing to make sense of. However, I would like to see the authors revise the manuscript to make it more clear what their research contributions have been.

For example:

1. Line 160. I got a little lost trying to keep separate the contributions of this paper and the comparisons to the literature in this paragraph. Given that this section is trying to help us understand the contributions of this paper, I think a couple of small tweaks could help make the distinctions more clear (suggestions in all caps)

"To be more specific, the MODEL'S heading estimation process gives rise to an error that systematically depends on object speed, with slow and fast moving objects causing a small error and an error peak for intermediate speeds, similar to findings of Dokka et al. (2019). Additionally, the MODEL'S direction of mis-estimation depends on whether the INDEPENDENTLY MOVING object maintains a fixed depth relative to the observer, a finding reported in various studies (Warren and Saunders, 1995; Royden and Hildreth, 1996; Li et al., 2018; Layton and Fajen, 2016b,d). THE MODEL'S OBJECT DETECTION PERFORMANCE depends on the deviation of the object flow from the background pattern, as Royden and colleagues found (Royden and Connors, 2010; Royden and Moore, 2012). While there is no research providing behavioral data in regard to object localization in optic flow fields, the model is able to successfully localize the independent source of motion solely based on flow velocities. Lastly, the object direction estimation is similar to human performance that shows that the perceived trajectory is consistent with scene-relative motion (Warren and Rushton, 2007, 2008, 2009).

2. It is a little difficult to follow the section on the computational model. I think it's because the section is missing a summary of their computational model at the beginning of the section. The introduction to that section lays out the equations for optic flow but that could be turned into a section entitled "Mathematical descriptions of optic flow".

- Additionally the section on residual surfaces waits til the final paragraph to explain why we should care about residual surfaces. And again we still have no idea why any of this might matter for their model.

- The section on the model dives into describing the purpose of each layer but feels quite disconnected from the first 3 sections in the methods.

3. The first two paragraphs in the results section (lines 634 - 668) seem like they are still describing the computational model construction and the simulation paradigm, rather than results. Could they be moved before the section break?

B. The only major scientific concern that I have, that I do not believe that the authors address, is how this model will be affected if the background dots are no longer uniformly distributed, but instead make up their own planar structures (more like the object). It's not obvious to me what will happen if the background has organized depth structures. I don't think the authors need to do more simulations (it would be cool but it seems outside the scope of this paper), but I do think it would be appropriate to address this issue in the discussion.

Minor issues:

1. I think it would be better not to abbreviate “independently moving object”. I understand that it is a phrase used a lot in this paper but it makes later sections more difficult to read. Part of this may be that in American English, IMO is a fairly common written slang abbreviation for “in my opinion” and I’m having to fight that definition as I read.
2. Line 45. “While suck radial patterns are often used in psychophysical studies, it is well known that eye-movements that occur during self-motion confound this simple structure by adding rotational components... Nonetheless studies with real as well as simulated eye movements showed that self-motion is still possible...”
It would be good to specify how this changes performance compared to the 1-2 degrees reported in the previous paragraph for simple radial flow patterns.
3. From line 504, you describe the dot field and object dots. I think it would be good to re-order this so that we have a full description of the dot cloud before you describe the object. Currently it jumps back and forth in a way that’s somewhat hard to follow.
4. [Just a comment] It’s interesting that the choice of the first layer operator spacing seems to matter for the function of the model.
5. (4.3.2. Object localization) — In the last paragraph you discuss that there are not appropriate behavior experiments to compare your results to. You might instead frame your work as a set of predictions for future behavioral work. Particularly if you think there are any surprising features of the model predictions.

Version 1:

Reviewer comments:

Reviewer #1

(Remarks to the Author)

In their revision, the authors have addressed my relatively minor comments quite satisfactorily. Their reorganization of the manuscript improves the reader's understanding their contribution: the review of previous models (illustrating the gap their model fills) now appears in the Introduction, the comparison of their simulations with previous human data comes in the Discussion, and the simulation details are presented in the final Methods.

This manuscript makes a very important contribution to the literature, providing a computational capstone to the heading and flow parsing problem. It should influence all future discussion of these issues. I enthusiastically recommend publication in its current form.

Reviewer #2

(Remarks to the Author)

The authors answered all of my comments clearly and made changes to the paper that I feel have improved it to the point where I am happy to recommend that the paper is accepted for publication.

I thank the authors for taking on board the comments from myself and the other reviewers, and I hope that they agree that this revised version makes an even stronger argument supporting the model.

Reviewer #3

(Remarks to the Author)

The authors have addressed the comments from my initial review.

This paper describes a new model that simultaneously solves two major problems of motion processing (heading estimation, and flow parsing) in the context of perception and action. The authors have done a great job of summarizing and synthesizing a large (and sometimes confusing) literature while also generating an interesting and novel computational approach for solving an important set of perceptual problems. This will be a valuable contribution to the visual and computational neuroscience literature.

General comments:

We thank the reviewers for their enthusiasm for the work and their detailed and helpful comments.

Following the suggestions and encouragement of the reviewers we performed some significant restructuring:

- 1) We streamlined the presentation of the simulation results by moving the comparison with human data to the discussion.
- 2) We moved the presentation of other models from the discussion to the introduction, to better illustrate the gap in the optic flow research landscape that our model fills..
- 3) We adjusted the structure to follow the journal's guidelines to present the model and simulations data in the results section and the simulation methods and mathematical details in a methods section at the end of the paper.

Below we provide detailed replies to the reviewers.

Reviewers' comments:

Reviewer #1 (Remarks to the Author):

Brief summary:

The first wave of research on optic flow focused on the 'rotation problem', how the visual system decomposes the flow field into components for observer translation (heading) and observer rotation. The most recent wave has focused on the problem of 'flow parsing', how the visual system separates the flow field into a global component due to self-motion (heading) and a local component due to object motion. Such flow patterns have been shown to produce specific human biases in both perceived heading and perceived object motion. In the present manuscript, the authors show that the leading model of the first problem (Heeger & Jepson, 1992; Lappe & Rauschecker, 1993) generalizes to the second problem – and naturally reproduces the observed pattern of human biases.

Overall impression:

The manuscript is a tour-de-force of modeling that succinctly explains a variety of human data and offers a coherent solution to two major problems of optic flow processing. First, the authors review the literature on flow parsing and summarize the puzzling pattern of biases that have been reported. Second, they extend their earlier neural model by cleverly splitting the output into a heading map and a moving object map. In a systematic set of simulations, they show that the model, with no further modification, explains the variety of human biases in both perceived heading and perceived object motion. The model provides a computationally unified and theoretically satisfying explanation of the flow parsing problem and the rotation problem, showing that they have a common root in flow field information. To top it off, the solution corresponds to Bayesian-optimal causal inference (Dokka, et al., 2019) without assuming Bayesian priors. This is an impressive piece of work that significantly advances our understanding of a central topic in human and machine vision.

Specific comments:

I only have a few comments, which can be addressed in a minor revision:

1. Please clarify the parameterization of object motion in the self-motion scenario (lines 530-552). I take it that observer translation toward the cloud is a vector with a magnitude $T = 2$ m/s in world coordinates. If the object is moving at $1 \cdot T$ “in the world,” then it would move in the same direction and at the same speed (2 m/s) as the observer, with no relative motion (‘neutral’). Yet the authors say that $1 \cdot T$ corresponds to the object ‘receding’ from the observer, whereas $0 \cdot T$ is ‘neutral’. What was the coordinate frame for object motion?

Reply: Yes, if the object is moving at $1 \cdot T$ “in the world,” then it would move in the same direction and at the same speed (2 m/s) as the observer, with no relative motion in depth between them. We see how this could be labelled as “neutral” but our terminology here refers to the motion in the world. In this view the object is moving backwards in depth to the same amount as the observer is moving forward. Thus we refer to the object as receding. We now specify this more clearly in the manuscript (lines 1422ff.).

2. A crucial aspect of the model is described very quickly and needs to be explained more clearly. Specifically, to what do the different residual surfaces in Layer 2 (Fig. 2, top) correspond? Based on Lines 401-411, I take it that each residual map represents the consistency of candidate headings all over the visual field with the output of a local operator for one receptive field. Is that correct? Please explain in the text.

Reply: Yes, you are right. We added an additional, summarizing part that aims to clarify the correspondence (lines 495ff.).

3. The flow parsing results strongly depend on the value of the activity threshold τ_1 for detecting a saddle point in Layer 3. Lines 648-668 say that τ_1 was selected so that a pure self-motion flow field was assigned to the heading estimation layer 90% of the time. Similarly, the detection of a moving object strongly depends on a second activity threshold τ_2 for a saddle point in the object estimation layer (call it Layer 4b). Lines 830-837 say that $\tau_2 = 1.5 \cdot \tau_1$. Please clarify why these values (90% and 1.5) were chosen. Given that there is no independent estimate of the values, would it be possible to fit them by reproducing the pattern of a subset of human data, and then to predict remaining human data, in some type of cross-validation?

Reply: Thank you for motioning this. In preliminary tests we have found that the results were rather robust over a range of choices for these thresholds. We now expanded our explanation for our initial choice of τ_1 (lines 611ff.) and discuss its influence on results (lines 637ff. for τ_1 and lines 776ff. for τ_2). Fitting the thresholds to human data does not seem possible at this stage as it would require additional data as, up to this point, studies only ever focused on singular aspects of optic flow processing, and the corresponding paradigms varied.

4. P. 13-17: The Results section begins to bog down in detail about here. The authors might be more selective about what they describe in the text to focus on the main points, and rely on figures more to express the details.

Reply: In response also to suggestions by Reviewer 2 we have streamlined the presentation of the simulation results and moved the comparison to human data to the discussion section.

Details:

- Line 303-307: This sentence could be clarified: “What the previous implementations have in common is that only the peak of the computed surface was taken as the final heading estimate, with no further attention paid to the distribution of residual values.”

Reply: We clarified that statement (lines 387ff.).

- Line 324 ff: The description of the residual surfaces in Figure 1 is a bit vague and difficult to understand, so it could be made more precise.

Reply: We expanded the description (lines 413ff.).

- Line 603: Should refer to Fig. 3F.

Reply: Corrected that mistake (line 1506).

- Figure 3: Fig. 3B and 3C are not particularly helpful because they are hard to interpret. Is this a top-down view of observer translation (red arrow) toward a cloud (white dots)? The cloud looks like a plane of dots receding in perspective, rather than a cloud in the viewing frustum. The streak of green dots looks like it lies on the plane, not like an object moving through (or occluding?) the cloud. I'd suggest that drawing a square cloud of dots (rather than a trapezoid) would help. Fig. 3E and 3F are impossible to interpret because the viewpoint is not clear – does it represent a frontal view like Fig. 3A and 3D, or a top-down view like Fig. 3B and 3C?

Reply: We also found this figure quite dense and are grateful for the helpful suggestions by the reviewer. In restructuring the paper we have moved a revised version of this figure to the detailed methods section at the end of the paper (now Figure 9) and replaced Fig. 3 with a more compact depiction that aims to show the differences between observer flow, object flow and combined flow and to illustrate the simulation conditions in an easier format (see also reviewer 2). We furthermore, in response to your comments reworked parts of the figure (now Fig 9) and the corresponding description. In detail:

- 1) We chose a different, less trivial, observer translation T as an example to emphasize that part of the object's movement is based on T (Panels b) and c)).
- 2) The object is now represented as a single bar, not as a collection of dots (Panels b) and c)).
- 3) The cloud of dots that form the static environment are now shown as a rectangular cloud, not as the rather misleading trapezoid (Panels b) and c)).
- 4) We added mentioning the viewpoint for the different types of panels.

Reviewer #2 (Remarks to the Author)

In their paper “Flow parsing as causal source separation: A computational model for concurrent retrieval of object and self-motion information from optic flow”, the authors generate just that; a model that allows for estimation of self-movement and object movement from optic flow concurrently. This is a mechanism that has been previously reported after behavioural investigation, but no model has yet, to my knowledge, modelled this behaviour. This modelling appears to closely match behaviour in heading and scene-relative object movement tasks, without assuming that one is necessary for the other

(another recent finding in the field). This model provides the underpinning mechanism that may be at play when we perform flow parsing which is an intriguing thought. The work is convincing, though I feel that a more convincing story could be told, as will be mentioned in my comments below. Perhaps some data showing that this model fits human data better than other previous models would strengthen the conclusions of the paper, but these are clearly, and with reason, beyond the scope of this paper and I would not be surprised if the authors are already looking into that for a subsequent publication. This work is a computational model, so it is infinitely repeatable with the provided codes, I thank the authors for that.

As mentioned above, I feel that there are structural changes that would help to give the reader a better understanding of the gap in the research that this model fills, while making the results section more digestible and allowing the discussion section to contain a more glowing review of the model. Alongside this, I have some points of intrigue that I would appreciate the authors' thoughts on. Please find more detailed comments about these below.

I should also point out that there were a few situations where the grammar could be improved.

Detailed Comments – Structure:

When reading section 5.2 “Comparison to other computational models”, I felt that there was little in the way of discussion or comparison between other models and your model (until the paragraph starting on line 1350). Instead, this felt like an introduction to the other computational models that exist and a well written unpacking of the gap in the research that your model fills. For this reason, I suggest that this section could be rebranded and used as part of the Introduction, in a section outlining current computational models of heading estimation and IMO movement estimation and identifying that, previous to yours, there is not a model that sufficiently explains heading estimation and IMO movement estimation concurrently.

This will leave a gap in the discussion section, which, I think can be filled by simplifying the results section. I feel that comparing the model to human performance (lines 738-743), making conclusions about the results (lines 809-820) etc. should form the basis of section 5.1 of the discussion section. Each subsection of the results section can be linked to in the discussion section e.g. “in section 4.3.1, our model was able to detect moving objects with a detection rate of x% at speed increases of y% and deviations of zdeg. This result fits with studies with human participants who showed...”. I think that this change will make the results section shorter and more direct with its presentation of what the model outputs, while allowing for clear description of how the model fits with human data in the discussion section.

I think that the discussion section containing a short recap of each result and then linking that directly to other research would provide a more glowing review of the model and what it achieves.

Reply: We took these ideas and the related comments of Reviewer #1 as inspiration for restructuring the manuscript. First, we streamlined the presentation of the simulation results by moving the comparison with human data to the discussion. Second, we moved the presentation of other models from the discussion to the introduction, to better illustrate the gap in the optic flow research landscape that our model fills. And, third, in following the journal's guidelines we moved the simulation methods and mathematical details to a methods section at the end of the paper.

In Results section 4.1, the first two paragraphs discuss the selection of the activity threshold T1 based on the percentage of residual surfaces that are assigned to the self-movement detection layer of the model when there is no IMO present. As this is “the first step in testing the model”, I feel that it should have its own section, perhaps section 4.1.1 “Selecting T1”. I would then denote the next section as section 4.1.2 “Speed Ratio”, or something similar.

Reply: We agree and added respective section breaks (lines 597ff.).

Detailed Comments – points of intrigue:

The third paragraph of section 4.1 discusses how changing the horizontal object speed affects the percentage of correct assignment, shown in Figure 4B. I am intrigued as to why the receding, neutral and semi-receding conditions produce similar percentage of correct assignment, while there is a big difference between those conditions and the approaching conditions. Is this because of how these conditions affect the speed ratio?

Reply: Indeed, this is due to the speed ratio for the different motion conditions, but also the direction deviation. Essentially, Combined flow is Observer flow - λ * Observer flow + Flow due to the horizontal component H. In detail:

- 1) (Semi-)Approaching condition: Combined flow is faster than the observer flow even if the horizontal movement speed H is 0, as λ is -1 or -0.5.
- 2) Neutral: Combined flow as fast as observer flow for H=0, but already faster for slow H, as $\lambda = 0$. Direction deviation is relatively small for slow H.
- 3) Receding: As the combined flow is basically only based on H, it is often slower than the observer flow for slow H. However, the direction deviation can be quite high, as the radial component is omitted (Combined flow is always vertical, no matter the object's retinal location).
- 4) Semi-Receding: Slightly worse than neutral and receding for slow H, as the speed of the combined flow is most likely slower than the observer flow while still carrying half of the radial component, which reduces the directional deviation.

This is what we believe the most likely causes. However, object placement and depth distribution of object and scene points can lead to cases not covered by this explanation. We expanded the discussion of the results to cover the differences in flow parsing quality (lines 1049ff.).

In fact, I feel that your description of the speed ratio could potentially benefit from a figure which explains what is meant by observer flow, object flow and combined flow, as I often found myself trying to imagine what each of these would be like in different scenarios.

Reply: We agree and have added a figure (Fig. 3) dedicated to showing the motion conditions and the differences and relations between the different flow types. Additionally, we use this figure to provide a better explanation for speed ratio and directional deviation in the results section. The former Fig. 3 has been moved to the methods section at the end of the paper, now Fig. 9 (see also comments of reviewer 1)

It also intrigues me that the object speed value whereby the parsing quality is effectively equivalent (37.5cm/s in Figure 4B) across the conditions, lines up quite nicely with the object speed value whereby the heading estimation, bias towards IMO location, object detection and object localisation measures (in Figure 5A and C-E) begin to be somewhat similar. Bias in IMO direction also appears to be similar above this object speed, though inverted for the receding conditions. I don't think this was mentioned in the text, but it could be an interesting finding if replicable with human participants (a critical speed value for consistent object motion perception with approaching/receding objects?)

Reply: The existence of such a critical speed in our results for which the estimations for different motion conditions align is a good observation, one we haven't appropriately described before. It is a consequence of the flow parsing quality for which performance also aligns across motion conditions for that speed. We added this to the discussion (lines 1043ff.).

In section 4.2.2 you simulate isolated IMOs which create largely similar results to when those IMOs aren't isolated. This makes sense as your model uses the flow across the whole receptive field for its processing. As I understand it, the reason behind isolating the IMO in Warren and Rushton (2009) was to remove the possibility of local motion processing (effectively processing the differential of the flow, rather than the flow itself). Could a version of your model be conceived where local motion processing was possible? How do you think this would change your results?

Reply: Indeed, our model does not rely on the type of local motion processing that Warren and Rushton rule out. We mention this now in the discussion section. We discuss why our model does not rely on such processes compared to other flow processing models (lines 1138ff.).

That being said, there are ways to restrict the computation of the subspace algorithm to only flow vectors that are close to each other (see Beintema, J. A. van den Berg, A. V., & Lappe, M. (2004). The structure of receptive fields for flow analysis and heading detection. In L. M Vaina, S. A. Beardsley, and S. Rushton, editors, *Optic Flow And Beyond*. Kluwer Academic Press, 1-24). How this would affect the flow parsing in the present model is an intriguing question but it would certainly not be consistent with Warren and Rushton (2009).

Reviewer #3 (Remarks to the Author):

Scherff & Lappe present a paper entitled "Flow parsing as causal source of separation: A computational model for concurrent retrieval of object and self-motion information from optic flow". This paper describes a new model that simultaneously solves two major problems of motion processing (heading estimation, and flow parsing) in the context of perception and action.

I genuinely enjoyed reading this paper. The authors have done a great job of summarizing and synthesizing a large (and sometimes confusing) literature while also generating an

interesting and novel computational approach for solving an important set of perceptual problems.

My major concerns are as follows:

A. One of the strengths of this paper is the clear manner in which it reviews and synthesizes the flow parsing literature. I think this is a great thing to do, particularly as the flow parsing literature can be quite confusing to make sense of. However, I would like to see the authors revise the manuscript to make it more clear what their research contributions have been.

Reply: Thank you for the encouragement. Indeed, also in response to similar suggestions from the other reviewers, we have restructured the manuscript to better separate the existing literature, our new model, its novel properties and capabilities, and the comparison to the existing literature.

First, we streamlined the presentation of the simulation results by moving the comparison with human data to the discussion. Second, we moved the presentation of other models from the discussion to the introduction, to better illustrate the gap in the optic flow research landscape that our model fills. And, third, in following the journal's guidelines we moved the simulation methods and mathematical details to a methods section at the end of the paper.

For example:

1. Line 160. I got a little lost trying to keep separate the contributions of this paper and the comparisons to the literature in this paragraph. Given that this section is trying to help us understand the contributions of this paper, I think a couple of small tweaks could help make the distinctions more clear (suggestions in all caps)

“To be more specific, the MODEL’S heading estimation process gives rise to an error that systematically depends on object speed, with slow and fast moving objects causing a small error and an error peak for intermediate speeds, similar to findings of Dokka et al. (2019). Additionally, the MODEL’S direction of mis-estimation depends on whether the INDEPENDENTLY MOVING object maintains a fixed depth relative to the observer, a finding reported in various studies (Warren and Saunders, 1995; Royden and Hildreth, 1996; Li et al., 2018; Layton and Fajen, 2016b,d). THE MODEL’S OBJECT DETECTION PERFORMANCE depends on the deviation of the object flow from the background pattern, as Royden and colleagues found (Royden and Connors, 2010; Royden and Moore, 2012). While there is no research providing behavioral data in regard to object localization in optic flow fields, the model is able to successfully localize the independent source of motion solely based on flow velocities. Lastly, the object direction estimation is similar to human performance that shows that the perceived trajectory is consistent with scene-relative motion (Warren and Rushton, 2007, 2008, 2009).

Reply: Thank you. We followed these suggestions (lines 318ff.)

2. It is a little difficult to follow the section on the computational model. I think it's because the section is missing a summary of their computational model at the beginning of the section. The introduction to that section lays out the equations for optic flow but that could be turned into a section entitled “Mathematical descriptions of optic flow”.

Reply: We hope that our restructuring has taken care of this. The results section now begins with a summary of the model and an explanation for why residual surfaces are

relevant. The mathematical descriptions of optic flow and the details of the subspace algorithm have been moved towards a method section at the end of the paper.

- Additionally the section on residual surfaces waits til the final paragraph to explain why we should care about residual surfaces. And again we still have no idea why any of this might matter for their model.

Reply: This is now presented at the start of the results section.

- The section on the model dives into describing the purpose of each layer but feels quite disconnected from the first 3 sections in the methods.

Reply: We hope due to the restructuring, that this is no longer an issue.

3. The first two paragraphs in the results section (lines 634 - 668) seem like they are still describing the computational model construction and the simulation paradigm, rather than results. Could they be moved before the section break?

Reply: Due to the restructuring in accordance with the journal guidelines these paragraphs now appear later in the results section, allowing for a better reading flow. Following a request by Reviewer #2 we transformed them into a section with the purpose of choosing the threshold.

B. The only major scientific concern that I have, that I do not believe that the authors address, is how this model will be affected if the background dots are no longer uniformly distributed, but instead make up their own planar structures (more like the object). It's not obvious to me what will happen if the background has organized depth structures. I don't think the authors need to do more simulations (it would be cool but it seems outside the scope of this paper), but I do think it would be appropriate to address this issue in the discussion.

Reply: We agree that the investigation of various types of environments would be important but beyond the scope of this paper. We added a paragraph to the discussion in which we briefly discuss other types of environmental depth structures and why they might not be too troublesome for the model (lines 1163ff.).

Minor issues:

1. I think it would be better not to abbreviate "independently moving object". I understand that it is a phrase used a lot in this paper but it makes later sections more difficult to read. Part of this may be that in American English, IMO is a fairly common written slang abbreviation for "in my opinion" and I'm having to fight that definition as I read.

Reply: We have replaced most instances of the abbreviation but keep a few in places where space is scarce, such as figure legends.

2. Line 45. "While suck radial patterns are often used in psychophysical studies, it is well known that eye-movements that occur during self-motion confound this simple structure by

adding rotational components... Nonetheless studies with real as well as simulated eye movements showed that self-motion is still possible...”

It would be good to specify how this changes performance compared to the 1-2 degrees reported in the previous paragraph for simple radial flow patterns.

Reply: We expended that part to report the performances in more detail (lines 51ff.).

3. From line 504, you describe the dot field and object dots. I think it would be good to re-order this so that we have a full description of the dot cloud before you describe the object. Currently it jumps back and forth in a way that's somewhat hard to follow.

Reply: We reworked that section lines(1404ff.).

4. [Just a comment] It's interesting that the choice of the first layer operator spacing seems to matter for the function of the model.

Reply: Indeed! Intuitively, it is understandable why the subspace algorithm shows some quirky behavior in such a case. If a candidate heading direction coincides with the location of one of the layer 1 operators, flow at that location could never be due to that specific translation direction. As the subspace algorithm is, simply speaking, a simultaneous orthogonality test for all flow vectors, such a null vector causes issues. Nonetheless, it would be unreasonable to expect human behavior to show a similar pattern. Hence, we adjusted the layer 1 operator placement to avoid that.

5. (4.3.2. Object localization) — In the last paragraph you discuss that there are not appropriate behavior experiments to compare your results to. You might instead frame your work as a set of predictions for future behavioral work. Particularly if you think there are any surprising features of the model predictions.

Reply: While the localization performance on its own might not provide interesting predictions (faster objects are easier to detect), we added some thoughts what overarching insights our model results offer and for which one might looking out when collecting new data (s. Comment to Reviewer #2) (lines 1043ff.).

Changes to figures:

Figure 1

Changed color of the heading direction indication to black.

Changed panel naming convention from A,B,C to a), b), c) (same for figures 4-9)

Figure 2

Changed color of the heading direction indication to black.

Figure 3

New figure to help illustrate the different types of flow, motion conditions and the flow metrics speed ratio and direction deviation

Figure 9

Change in color choice for various parts to align with previous figures (b,c,d,e,f).

Added translation direction indicator (a,f).

Changed observer translation direction, object and environment representation for easier to understand explanations (b,c)

Reviewer answers to manuscript revision and author comments:

As no further requests from any reviewer are pending, we want to thank the reviewers again for their contributions to our work. Their comments were fair and understandable, and their suggestions were very helpful in reshaping the manuscript and clarifying some sections. We absolutely agree that the revised version of the manuscript is a clear improvement over the initial version.

Reviewer #1 (Remarks to the Author):

In their revision, the authors have addressed my relatively minor comments quite satisfactorily. Their reorganization of the manuscript improves the reader's understanding their contribution: the review of previous models (illustrating the gap their model fills) now appears in the Introduction, the comparison of their simulations with previous human data comes in the Discussion, and the simulation details are presented in the final Methods. This manuscript makes a very important contribution to the literature, providing a computational capstone to the heading and flow parsing problem. It should influence all future discussion of these issues. I enthusiastically recommend publication in its current form.

Reviewer #2 (Remarks to the Author):

The authors answered all of my comments clearly and made changes to the paper that I feel have improved it to the point where I am happy to recommend that the paper is accepted for publication. I thank the authors for taking on board the comments from myself and the other reviewers, and I hope that they agree that this revised version makes an even stronger argument supporting the model.

Reviewer #3 (Remarks to the Author):

The authors have addressed the comments from my initial review.

This paper describes a new model that simultaneously solves two major problems of motion processing (heading estimation, and flow parsing) in the context of perception and action. The authors have done a great job of summarizing and synthesizing a large (and sometimes confusing) literature while also generating an interesting and novel computational approach for solving an important set of perceptual problems. This will be a valuable contribution to the visual and computational neuroscience literature.

Initial review: Reviewer comments and author responses

General comments:

We thank the reviewers for their enthusiasm for the work and their detailed and helpful comments.

Following the suggestions and encouragement of the reviewers we performed some significant restructuring:

- 1) We streamlined the presentation of the simulation results by moving the comparison with human data to the discussion.
- 2) We moved the presentation of other models from the discussion to the introduction, to better illustrate the gap in the optic flow research landscape that our model fills..
- 3) We adjusted the structure to follow the journal's guidelines to present the model and simulations data in the results section and the simulation methods and mathematical details in a methods section at the end of the paper.

Below we provide detailed replies to the reviewers.

Reviewers' comments:

Reviewer #1 (Remarks to the Author):

Brief summary:

The first wave of research on optic flow focused on the 'rotation problem', how the visual system decomposes the flow field into components for observer translation (heading) and observer rotation. The most recent wave has focused on the problem of 'flow parsing', how the visual system separates the flow field into a global component due to self-motion (heading) and a local component due to object motion. Such flow patterns have been shown to produce specific human biases in both perceived heading and perceived object motion. In the present manuscript, the authors show that the leading model of the first problem (Heeger & Jepson, 1992; Lappe & Rauschecker, 1993) generalizes to the second problem – and naturally reproduces the observed pattern of human biases.

Overall impression:

The manuscript is a tour-de-force of modeling that succinctly explains a variety of human data and offers a coherent solution to two major problems of optic flow processing. First, the authors review the literature on flow parsing and summarize the puzzling pattern of biases that have been reported. Second, they extend their earlier neural model by cleverly splitting the output into a heading map and a moving object map. In a systematic set of simulations, they show that the model, with no further modification, explains the variety of human biases in both perceived heading and perceived object motion. The model provides a computationally unified and theoretically satisfying explanation of the flow parsing problem and the rotation problem, showing that they have a common root in flow field information. To top it off, the solution corresponds to Bayesian-optimal causal inference (Dokka, et al., 2019) without assuming Bayesian priors. This is an impressive piece of work that significantly advances our understanding of a central topic in human and machine vision.

Specific comments:

I only have a few comments, which can be addressed in a minor revision:

1. Please clarify the parameterization of object motion in the self-motion scenario (lines 530-552). I take it that observer translation toward the cloud is a vector with a magnitude $T = 2$ m/s in world coordinates. If the object is moving at $1 \cdot T$ “in the world,” then it would move in the same direction and at the same speed (2 m/s) as the observer, with no relative motion (‘neutral’). Yet the authors say that $1 \cdot T$ corresponds to the object ‘receding’ from the observer, whereas $0 \cdot T$ is ‘neutral’. What was the coordinate frame for object motion?

Reply: Yes, if the object is moving at $1 \cdot T$ “in the world,” then it would move in the same direction and at the same speed (2 m/s) as the observer, with no relative motion in depth between them. We see how this could be labelled as “neutral” but our terminology here refers to the motion in the world. In this view the object is moving backwards in depth to the same amount as the observer is moving forward. Thus we refer to the object as receding. We now specify this more clearly in the manuscript (lines 1422ff.).

2. A crucial aspect of the model is described very quickly and needs to be explained more clearly. Specifically, to what do the different residual surfaces in Layer 2 (Fig. 2, top) correspond? Based on Lines 401-411, I take it that each residual map represents the consistency of candidate headings all over the visual field with the output of a local operator for one receptive field. Is that correct? Please explain in the text.

Reply: Yes, you are right. We added an additional, summarizing part that aims to clarify the correspondence (lines 495ff.).

3. The flow parsing results strongly depend on the value of the activity threshold τ_1 for detecting a saddle point in Layer 3. Lines 648-668 say that τ_1 was selected so that a pure self-motion flow field was assigned to the heading estimation layer 90% of the time. Similarly, the detection of a moving object strongly depends on a second activity threshold τ_2 for a saddle point in the object estimation layer (call it Layer 4b). Lines 830-837 say that $\tau_2 = 1.5 \cdot \tau_1$. Please clarify why these values (90% and 1.5) were chosen. Given that there is no independent estimate of the values, would it be possible to fit them by reproducing the pattern of a subset of human data, and then to predict remaining human data, in some type of cross-validation?

Reply: Thank you for motioning this. In preliminary tests we have found that the results were rather robust over a range of choices for these thresholds. We now expanded our explanation for our initial choice of τ_1 (lines 611ff.) and discuss its influence on results (lines 637ff. for τ_1 and lines 776ff. for τ_2). Fitting the thresholds to human data does not seem possible at this stage as it would require additional data as, up to this point, studies only ever focused on singular aspects of optic flow processing, and the corresponding paradigms varied.

4. P. 13-17: The Results section begins to bog down in detail about here. The authors might be more selective about what they describe in the text to focus on the main points, and rely on figures more to express the details.

Reply: In response also to suggestions by Reviewer 2 we have streamlined the presentation of the simulation results and moved the comparison to human data to the discussion section.

Details:

- Line 303-307: This sentence could be clarified: “What the previous implementations have in common is that only the peak of the computed surface was taken as the final heading estimate, with no further attention paid to the distribution of residual values.”

Reply: We clarified that statement (lines 387ff.).

- Line 324 ff: The description of the residual surfaces in Figure 1 is a bit vague and difficult to understand, so it could be made more precise.

Reply: We expanded the description (lines 413ff.).

- Line 603: Should refer to Fig. 3F.

Reply: Corrected that mistake (line 1506).

- Figure 3: Fig. 3B and 3C are not particularly helpful because they are hard to interpret. Is this a top-down view of observer translation (red arrow) toward a cloud (white dots)? The cloud looks like a plane of dots receding in perspective, rather than a cloud in the viewing frustum. The streak of green dots looks like it lies on the plane, not like an object moving through (or occluding?) the cloud. I'd suggest that drawing a square cloud of dots (rather than a trapezoid) would help. Fig. 3E and 3F are impossible to interpret because the viewpoint is not clear – does it represent a frontal view like Fig. 3A and 3D, or a top-down view like Fig. 3B and 3C?

Reply: We also found this figure quite dense and are grateful for the helpful suggestions by the reviewer. In restructuring the paper we have moved a revised version of this figure to the detailed methods section at the end of the paper (now Figure 9) and replaced Fig. 3 with a more compact depiction that aims to show the differences between observer flow, object flow and combined flow and to illustrate the simulation conditions in an easier format (see also reviewer 2). We furthermore, in response to your comments reworked parts of the figure (now Fig 9) and the corresponding description. In detail:

- 1) We chose a different, less trivial, observer translation T as an example to emphasize that part of the object's movement is based on T (Panels b) and c)).
- 2) The object is now represented as a single bar, not as a collection of dots (Panels b) and c)).
- 3) The cloud of dots that form the static environment are now shown as a rectangular cloud, not as the rather misleading trapezoid (Panels b) and c)).
- 4) We added mentioning the viewpoint for the different types of panels.

Reviewer #2 (Remarks to the Author)

In their paper “Flow parsing as causal source separation: A computational model for concurrent retrieval of object and self-motion information from optic flow”, the authors generate just that; a model that allows for estimation of self-movement and object movement from optic flow concurrently. This is a mechanism that has been previously reported after behavioural investigation, but no model has yet, to my knowledge, modelled this behaviour. This modelling appears to closely match behaviour in heading and scene-relative object movement tasks, without assuming that one is necessary for the other

(another recent finding in the field). This model provides the underpinning mechanism that may be at play when we perform flow parsing which is an intriguing thought. The work is convincing, though I feel that a more convincing story could be told, as will be mentioned in my comments below. Perhaps some data showing that this model fits human data better than other previous models would strengthen the conclusions of the paper, but these are clearly, and with reason, beyond the scope of this paper and I would not be surprised if the authors are already looking into that for a subsequent publication. This work is a computational model, so it is infinitely repeatable with the provided codes, I thank the authors for that.

As mentioned above, I feel that there are structural changes that would help to give the reader a better understanding of the gap in the research that this model fills, while making the results section more digestible and allowing the discussion section to contain a more glowing review of the model. Alongside this, I have some points of intrigue that I would appreciate the authors' thoughts on. Please find more detailed comments about these below.

I should also point out that there were a few situations where the grammar could be improved.

Detailed Comments – Structure:

When reading section 5.2 “Comparison to other computational models”, I felt that there was little in the way of discussion or comparison between other models and your model (until the paragraph starting on line 1350). Instead, this felt like an introduction to the other computational models that exist and a well written unpacking of the gap in the research that your model fills. For this reason, I suggest that this section could be rebranded and used as part of the Introduction, in a section outlining current computational models of heading estimation and IMO movement estimation and identifying that, previous to yours, there is not a model that sufficiently explains heading estimation and IMO movement estimation concurrently.

This will leave a gap in the discussion section, which, I think can be filled by simplifying the results section. I feel that comparing the model to human performance (lines 738-743), making conclusions about the results (lines 809-820) etc. should form the basis of section 5.1 of the discussion section. Each subsection of the results section can be linked to in the discussion section e.g. “in section 4.3.1, our model was able to detect moving objects with a detection rate of x% at speed increases of y% and deviations of zdeg. This result fits with studies with human participants who showed...”. I think that this change will make the results section shorter and more direct with its presentation of what the model outputs, while allowing for clear description of how the model fits with human data in the discussion section.

I think that the discussion section containing a short recap of each result and then linking that directly to other research would provide a more glowing review of the model and what it achieves.

Reply: We took these ideas and the related comments of Reviewer #1 as inspiration for restructuring the manuscript. First, we streamlined the presentation of the simulation results by moving the comparison with human data to the discussion. Second, we moved the presentation of other models from the discussion to the introduction, to better illustrate the gap in the optic flow research landscape that our model fills. And, third, in following the journal's guidelines we moved the simulation methods and mathematical details to a methods section at the end of the paper.

In Results section 4.1, the first two paragraphs discuss the selection of the activity threshold T1 based on the percentage of residual surfaces that are assigned to the self-movement detection layer of the model when there is no IMO present. As this is “the first step in testing the model”, I feel that it should have its own section, perhaps section 4.1.1 “Selecting T1”. I would then denote the next section as section 4.1.2 “Speed Ratio”, or something similar.

Reply: We agree and added respective section breaks (lines 597ff.).

Detailed Comments – points of intrigue:

The third paragraph of section 4.1 discusses how changing the horizontal object speed affects the percentage of correct assignment, shown in Figure 4B. I am intrigued as to why the receding, neutral and semi-receding conditions produce similar percentage of correct assignment, while there is a big difference between those conditions and the approaching conditions. Is this because of how these conditions affect the speed ratio?

Reply: Indeed, this is due to the speed ratio for the different motion conditions, but also the direction deviation. Essentially, Combined flow is Observer flow - λ * Observer flow + Flow due to the horizontal component H. In detail:

- 1) (Semi-)Approaching condition: Combined flow is faster than the observer flow even if the horizontal movement speed H is 0, as λ is -1 or -0.5.
- 2) Neutral: Combined flow as fast as observer flow for H=0, but already faster for slow H, as $\lambda = 0$. Direction deviation is relatively small for slow H.
- 3) Receding: As the combined flow is basically only based on H, it is often slower than the observer flow for slow H. However, the direction deviation can be quite high, as the radial component is omitted (Combined flow is always vertical, no matter the object's retinal location).
- 4) Semi-Receding: Slightly worse than neutral and receding for slow H, as the speed of the combined flow is most likely slower than the observer flow while still carrying half of the radial component, which reduces the directional deviation.

This is what we believe the most likely causes. However, object placement and depth distribution of object and scene points can lead to cases not covered by this explanation. We expanded the discussion of the results to cover the differences in flow parsing quality (lines 1049ff.).

In fact, I feel that your description of the speed ratio could potentially benefit from a figure which explains what is meant by observer flow, object flow and combined flow, as I often found myself trying to imagine what each of these would be like in different scenarios.

Reply: We agree and have added a figure (Fig. 3) dedicated to showing the motion conditions and the differences and relations between the different flow types. Additionally, we use this figure to provide a better explanation for speed ratio and directional deviation in the results section. The former Fig. 3 has been moved to the methods section at the end of the paper, now Fig. 9 (see also comments of reviewer 1)

It also intrigues me that the object speed value whereby the parsing quality is effectively equivalent (37.5cm/s in Figure 4B) across the conditions, lines up quite nicely with the object speed value whereby the heading estimation, bias towards IMO location, object detection and object localisation measures (in Figure 5A and C-E) begin to be somewhat similar. Bias in IMO direction also appears to be similar above this object speed, though inverted for the receding conditions. I don't think this was mentioned in the text, but it could be an interesting finding if replicable with human participants (a critical speed value for consistent object motion perception with approaching/receding objects?)

Reply: The existence of such a critical speed in our results for which the estimations for different motion conditions align is a good observation, one we haven't appropriately described before. It is a consequence of the flow parsing quality for which performance also aligns across motion conditions for that speed. We added this to the discussion (lines 1043ff.).

In section 4.2.2 you simulate isolated IMOs which create largely similar results to when those IMOs aren't isolated. This makes sense as your model uses the flow across the whole receptive field for its processing. As I understand it, the reason behind isolating the IMO in Warren and Rushton (2009) was to remove the possibility of local motion processing (effectively processing the differential of the flow, rather than the flow itself). Could a version of your model be conceived where local motion processing was possible? How do you think this would change your results?

Reply: Indeed, our model does not rely on the type of local motion processing that Warren and Rushton rule out. We mention this now in the discussion section. We discuss why our model does not rely on such processes compared to other flow processing models (lines 1138ff.).

That being said, there are ways to restrict the computation of the subspace algorithm to only flow vectors that are close to each other (see Beintema, J. A. van den Berg, A. V., & Lappe, M. (2004). The structure of receptive fields for flow analysis and heading detection. In L. M Vaina, S. A. Beardsley, and S. Rushton, editors, *Optic Flow And Beyond*. Kluwer Academic Press, 1-24). How this would affect the flow parsing in the present model is an intriguing question but it would certainly not be consistent with Warren and Rushton (2009).

Reviewer #3 (Remarks to the Author):

Scherff & Lappe present a paper entitled "Flow parsing as causal source of separation: A computational model for concurrent retrieval of object and self-motion information from optic flow". This paper describes a new model that simultaneously solves two major problems of motion processing (heading estimation, and flow parsing) in the context of perception and action.

I genuinely enjoyed reading this paper. The authors have done a great job of summarizing and synthesizing a large (and sometimes confusing) literature while also generating an

interesting and novel computational approach for solving an important set of perceptual problems.

My major concerns are as follows:

A. One of the strengths of this paper is the clear manner in which it reviews and synthesizes the flow parsing literature. I think this is a great thing to do, particularly as the flow parsing literature can be quite confusing to make sense of. However, I would like to see the authors revise the manuscript to make it more clear what their research contributions have been.

Reply: Thank you for the encouragement. Indeed, also in response to similar suggestions from the other reviewers, we have restructured the manuscript to better separate the existing literature, our new model, its novel properties and capabilities, and the comparison to the existing literature.

First, we streamlined the presentation of the simulation results by moving the comparison with human data to the discussion. Second, we moved the presentation of other models from the discussion to the introduction, to better illustrate the gap in the optic flow research landscape that our model fills. And, third, in following the journal's guidelines we moved the simulation methods and mathematical details to a methods section at the end of the paper.

For example:

1. Line 160. I got a little lost trying to keep separate the contributions of this paper and the comparisons to the literature in this paragraph. Given that this section is trying to help us understand the contributions of this paper, I think a couple of small tweaks could help make the distinctions more clear (suggestions in all caps)

“To be more specific, the MODEL’S heading estimation process gives rise to an error that systematically depends on object speed, with slow and fast moving objects causing a small error and an error peak for intermediate speeds, similar to findings of Dokka et al. (2019). Additionally, the MODEL’S direction of mis-estimation depends on whether the INDEPENDENTLY MOVING object maintains a fixed depth relative to the observer, a finding reported in various studies (Warren and Saunders, 1995; Royden and Hildreth, 1996; Li et al., 2018; Layton and Fajen, 2016b,d). THE MODEL’S OBJECT DETECTION PERFORMANCE depends on the deviation of the object flow from the background pattern, as Royden and colleagues found (Royden and Connors, 2010; Royden and Moore, 2012). While there is no research providing behavioral data in regard to object localization in optic flow fields, the model is able to successfully localize the independent source of motion solely based on flow velocities. Lastly, the object direction estimation is similar to human performance that shows that the perceived trajectory is consistent with scene-relative motion (Warren and Rushton, 2007, 2008, 2009).

Reply: Thank you. We followed these suggestions (lines 318ff.)

2. It is a little difficult to follow the section on the computational model. I think it's because the section is missing a summary of their computational model at the beginning of the section. The introduction to that section lays out the equations for optic flow but that could be turned into a section entitled “Mathematical descriptions of optic flow”.

Reply: We hope that our restructuring has taken care of this. The results section now begins with a summary of the model and an explanation for why residual surfaces are

relevant. The mathematical descriptions of optic flow and the details of the subspace algorithm have been moved towards a method section at the end of the paper.

- Additionally the section on residual surfaces waits til the final paragraph to explain why we should care about residual surfaces. And again we still have no idea why any of this might matter for their model.

Reply: This is now presented at the start of the results section.

- The section on the model dives into describing the purpose of each layer but feels quite disconnected from the first 3 sections in the methods.

Reply: We hope due to the restructuring, that this is no longer an issue.

3. The first two paragraphs in the results section (lines 634 - 668) seem like they are still describing the computational model construction and the simulation paradigm, rather than results. Could they be moved before the section break?

Reply: Due to the restructuring in accordance with the journal guidelines these paragraphs now appear later in the results section, allowing for a better reading flow. Following a request by Reviewer #2 we transformed them into a section with the purpose of choosing the threshold.

B. The only major scientific concern that I have, that I do not believe that the authors address, is how this model will be affected if the background dots are no longer uniformly distributed, but instead make up their own planar structures (more like the object). It's not obvious to me what will happen if the background has organized depth structures. I don't think the authors need to do more simulations (it would be cool but it seems outside the scope of this paper), but I do think it would be appropriate to address this issue in the discussion.

Reply: We agree that the investigation of various types of environments would be important but beyond the scope of this paper. We added a paragraph to the discussion in which we briefly discuss other types of environmental depth structures and why they might not be too troublesome for the model (lines 1163ff.).

Minor issues:

1. I think it would be better not to abbreviate "independently moving object". I understand that it is a phrase used a lot in this paper but it makes later sections more difficult to read. Part of this may be that in American English, IMO is a fairly common written slang abbreviation for "in my opinion" and I'm having to fight that definition as I read.

Reply: We have replaced most instances of the abbreviation but keep a few in places where space is scarce, such as figure legends.

2. Line 45. "While suck radial patterns are often used in psychophysical studies, it is well known that eye-movements that occur during self-motion confound this simple structure by

adding rotational components... Nonetheless studies with real as well as simulated eye movements showed that self-motion is still possible...”

It would be good to specify how this changes performance compared to the 1-2 degrees reported in the previous paragraph for simple radial flow patterns.

Reply: We expended that part to report the performances in more detail (lines 51ff.).

3. From line 504, you describe the dot field and object dots. I think it would be good to re-order this so that we have a full description of the dot cloud before you describe the object. Currently it jumps back and forth in a way that's somewhat hard to follow.

Reply: We reworked that section lines(1404ff.).

4. [Just a comment] It's interesting that the choice of the first layer operator spacing seems to matter for the function of the model.

Reply: Indeed! Intuitively, it is understandable why the subspace algorithm shows some quirky behavior in such a case. If a candidate heading direction coincides with the location of one of the layer 1 operators, flow at that location could never be due to that specific translation direction. As the subspace algorithm is, simply speaking, a simultaneous orthogonality test for all flow vectors, such a null vector causes issues. Nonetheless, it would be unreasonable to expect human behavior to show a similar pattern. Hence, we adjusted the layer 1 operator placement to avoid that.

5. (4.3.2. Object localization) — In the last paragraph you discuss that there are not appropriate behavior experiments to compare your results to. You might instead frame your work as a set of predictions for future behavioral work. Particularly if you think there are any surprising features of the model predictions.

Reply: While the localization performance on its own might not provide interesting predictions (faster objects are easier to detect), we added some thoughts what overarching insights our model results offer and for which one might looking out when collecting new data (s. Comment to Reviewer #2) (lines 1043ff.).

Changes to figures:

Figure 1

Changed color of the heading direction indication to black.

Changed panel naming convention from A,B,C to a), b), c) (same for figures 4-9)

Figure 2

Changed color of the heading direction indication to black.

Figure 3

New figure to help illustrate the different types of flow, motion conditions and the flow metrics speed ratio and direction deviation

Figure 9

Change in color choice for various parts to align with previous figures (b,c,d,e,f).

Added translation direction indicator (a,f).

Changed observer translation direction, object and environment representation for easier to understand explanations (b,c)